# Overcoming the numerical challenges owing to rapid ductile localization with DEDLoc (version 1.0.0)

Arne Spang[1], Marcel Thielmann[1,2], Casper Pranger[3], Albert de Montserrat[4], and Ludovic Räss[5]

[1]Bayerisches Geoinstitut, Universität Bayreuth, Universitätsstraße 30, 95447 Bayreuth, Germany
[2]Institut für Geowissenschaften, Universität Bonn, Meckenheimer Allee 176, 53115 Bonn, Germany
[3]Department für Geo- und Umweltwissenschaften, Ludwig-Maximilians-Universität München, Theresienstraße 41, 80333 München, Germany
[4]Department of Earth Sciences, ETH Zürich, Sonneggstrasse 5, 8092 Zürich, Switzerland
[5]Swiss Geocomputing Centre, Faculty of Geosciences and Environment, University of Lausanne, Lausanne Switzerland

**Correspondence:** Arne Spang (arne.spang@uni-bayreuth.de)

**Abstract.** Strain localization is among the most challenging mechanical phenomena for computational Earth sciences. Accurately capturing it is difficult because strain localization initiates spontaneously, is self-accelerating, and its characteristic length and time scales are typically significantly smaller than the spatial and temporal resolutions of the model. This results in an undesirable dependence of the model behavior on numerical parameters and comes at a large computational cost. Strain localization is most commonly associated with brittle failure, but processes such as thermal runaway can also result in rapid ductile localization. Here, we present a numerical model to investigate thermal runaway, and propose strategies to overcome the challenges associated with resolving rapid localization: (i) adaptive time stepping; (ii) adaptive rescaling; (iii) viscosity regularization; and (iv) gradient regularization. We demonstrate the effect of these strategies in one- and two-dimensional models. We rely on the accelerated pseudo-transient method to solve the governing equations and use graphics processing units to accelerate two-dimensional computations. Our adaptive time stepping strategy allows us to accurately capture spontaneous and rapid stress release during thermal runaway while reducing time steps by more than ten orders of magnitude. Adaptive rescaling further reduces rounding errors and the number of required iterations by two orders of magnitude. Viscosity regularization and gradient regularization enable us to mitigate resolution dependencies but may differently impact the physical response of the model. Viscosity regularization results in lower slip velocities, whereas gradient regularization results in lower temperatures and broader shear zones.

## 1 Introduction

Strain localization is a mechanism that focuses distributed deformation into a narrow zone (shear band or shear zone) which allows relatively stiff blocks to move past each other without significant internal deformation. It is a critical component of solid deformation that can be observed on any scale and in almost any material (Poirier, 1980; De Borst et al., 1993; Desrues et al., 2007; Antolovich and Armstrong, 2014; Weidner and Biermann, 2021). Strain localization governs tectonic processes such as

subduction (e.g., Auzemery et al., 2020) and orogenesis (e.g., Roy et al., 2016), as well as hazards like landslides (e.g., Darve and Laouafa, 2000) and earthquakes (e.g., Barras and Brantut, 2025).

In Geodynamics, modeling strain localization accurately and reproducibly remains inherently challenging due to the large differences in involved scales. A model has to cover the km-scale geological setting which evolves on time scales of kyr as well as the mm-scale localized shear zone which may operate on time scales of seconds. Furthermore, the self-feeding character of strain localization usually results in a lack of a finite length and time scale (De Borst et al., 1993; Iordache and Willam, 1998; Gerolymatou et al., 2024). As a consequence, the model behavior becomes dependent on numerical parameters such as spatial and temporal resolution and fails to accurately capture strain localization. A plate-scale model ($\sim 10^6$ m) is likely to overestimate the width of a shear zone due to its coarse spatial resolution. A grain-scale model ($\sim 10^{-3}$ m) might underestimate shear zone width if its domain is too small to cover the relevant geological context. Another challenge in resolving spontaneous localization is the broad spectrum of values that must be covered with sufficient numerical accuracy.

In the Earth's lithosphere, strain localization predominantly occurs via brittle failure where the stress in a rock unit exceeds its strength, and it breaks into separate blocks that slide on a fault. With increasing depth and lithostatic pressure, the brittle strength of rocks increases linearly (Drucker and Prager, 1952; Byerlee, 1978), while increasing temperatures promote ductile deformation. If brittle failure were the only mechanism to localize deformation, this would suggest that highly localized deformation should be limited to less than about 100 km depth. However, the occurrence of deep earthquakes, reaching depths of about 660 km (Turner, 1922; Wadati, 1928; Leith and Sharpe, 1936), demonstrates that strong strain localization and rapid slip can also occur under conditions that favor ductile deformation.

In ductile localization, there is no complete loss of cohesion (i.e., breaking). Instead, an area of the material weakens to the point where it can accommodate most or all of the large scale deformation (Poirier, 1980; Burg, 1999; Katz et al., 2006). One mechanism proposed to facilitate ductile localization is thermal runaway (Gruntfest, 1963; Ogawa, 1987). This process, illustrated in Fig. 1a, describes a feedback cycle that includes deformation, shear heating (or viscous dissipation), temperature-dependent rheology, and localization. Once deformation begins to localize within a weak inclusion embedded in a stronger matrix (Fig. 2), shear heating causes the temperature in the inclusion to rise more rapidly, thereby locally reducing the viscosity and further enhancing localization. This feedback loop can result in catastrophic strength reduction, a surge in temperature, rapid stress release, and highly localized slip (e.g., Kameyama et al., 1999; Kelemen and Hirth, 2007; Thielmann et al., 2015). Thermal diffusion can stop this feedback loop if sufficient heat is transferred from the shear zone to the surrounding host rock, which prevents further increase in the viscosity contrast between the units (Braeck et al., 2009; Thielmann, 2018; Spang et al., 2024).

In Spang et al. (2024), we captured the dynamics of thermal runaway using a one-dimensional, visco-elastic thermomechanical simple shear model, which predicts the temporal evolution of stress and temperature within an evolving shear zone (Fig. 1b). The model evolves through five distinct stages: (i) elastic loading, during which deviatoric stress increases linearly while temperature remains constant; (ii) steady-state viscous creep, dominated by low-temperature plasticity (LTP), where stress remains nearly constant and temperature increases steadily; (iii) thermal runaway, in which deformation localizes into a narrowing slip zone dominated by dislocation creep, leading to a significant stress drop and an exponential increase in tempera-

ture; (iv) post-runaway loading, characterized by linear stress increase as heat diffuses from the shear zone into the surrounding host rock; and (v) post-runaway creep, where temperature is large enough for dislocation creep to gradually relax stress as the system transitions into a stable sliding regime.

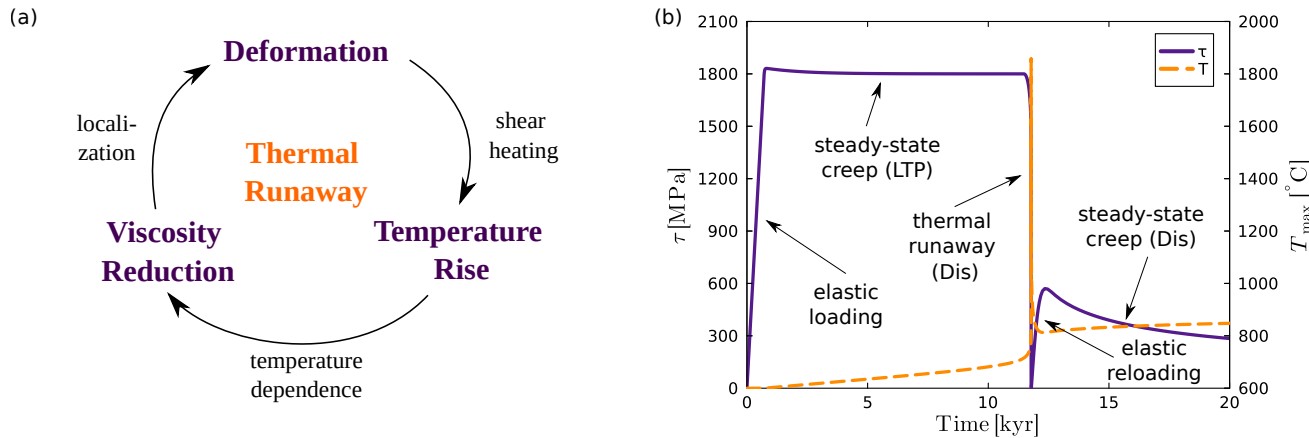

**Figure 1.** Illustration of thermal runaway. (a) Feedback cycle of processes that combine to make thermal runaway. (b) Temporal evolution of deviatoric stress (purple) and maximum temperature (orange). Arrows indicate different stages of the model evolution. LTP and Dis are short for low-temperature plasticity and dislocation creep respectively, and indicate the dominant deformation mechanism of the stages.

The transient and nonlinear runaway phase presents several challenges that thermomechanical models must overcome to achieve an accurate numerical solution: (i) spontaneous initiation; (ii) poor nonlinear solver convergence; and (iii) mesh-dependent results. Modeling brittle failure/localization suffers from similar issues (e.g., Spiegelman et al., 2016; Duretz et al., 2020).

In this study, we present and discuss the one- and two-dimensional (1D and 2D) models we used to capture spontaneous ductile shear localization. We incorporate a visco-elastic, composite rheology and utilize the accelerated pseudo-transient (APT) method (Frankel, 1950; Räss et al., 2022; Alkhimenkov and Podladchikov, 2024) to solve the governing system of equations. We then focus on the numerical challenges associated with rapid localization and describe our strategies to overcome them: (i) adaptive time stepping; (ii) adaptive rescaling; (iii) viscosity regularization; (iv) gradient regularization; and (v) enforcing viscosity convergence. Readers interested in the application of these models are referred to Spang et al. (2024) and Spang et al. (2025a) for the 1D and 2D cases, respectively.

## 2 Methods

### 2.1 Governing equations

To capture rapid ductile shear localization, we consider a system of coupled thermomechanical equations governing the conservation of momentum, mass, and energy:

$$\frac{\partial \tau_{ij}}{\partial x_j} - \frac{\partial P}{\partial x_i} = 0 \; , \tag{1}$$

$$\frac{1}{\rho}\frac{\partial \rho}{\partial t} = -\frac{\partial v_i}{\partial x_i} \; , \tag{2}$$

$$\rho C_p \frac{\partial T}{\partial t} = \frac{\partial}{\partial x_i}\left( k\frac{\partial T}{\partial x_i}\right) + \tau_{ij}\dot{\varepsilon}_{ij}^{\mathrm{vi}} \; , \tag{3}$$

where $\tau_{ij}$ is the Cauchy stress deviator, $x_i$ denotes the Cartesian coordinates, $P$ is pressure (positive in compression), $\rho$ is density, $t$ is time, $v_i$ is the velocity vector, $C_p$ is specific heat capacity, $T$ is temperature, $k$ is thermal conductivity, and $\dot{\varepsilon}_{ij}^{\mathrm{vi}}$ is the viscous component of the deviatoric strain rate. For simplicity, we neglect the inertial terms and body forces (i.e. gravity) from Eq. (1) as well as adiabatic and radiogenic heating from Eq. (3). The last term of Eq. (3) describes energy from viscous dissipation and it is entirely partitioned into shear heating. These simplifications are discussed in Sect. 6.

The conservation equations are augmented by a constitutive relation for bulk compressibility:

$$\frac{1}{K_{\mathrm{b}}}\frac{\partial P}{\partial t} = -\frac{\partial v_i}{\partial x_i} \; , \tag{4}$$

where $K_{\mathrm{b}}$ is the bulk modulus. For simplicity, we neglect thermal expansion from Eq. (4). Combining equations (2) and (4), and integrating the changes in pressure and density yields the equation of state for density:

$$\rho = \rho_{\mathrm{ref}} \exp\left(\frac{P - P_{\mathrm{ref}}}{K_{\mathrm{b}}}\right) \; , \tag{5}$$

where $\rho_{\mathrm{ref}}$ and $P_{\mathrm{ref}}$ are the reference density and pressure at atmospheric conditions respectively (Gerya, 2019, p. 26).

The deviatoric strain rate is defined as:

$$\dot{\varepsilon}_{ij} = \frac{1}{2}\left(\frac{\partial v_i}{\partial x_j} + \frac{\partial v_j}{\partial x_i}\right) - \frac{1}{3}\frac{\partial v_k}{\partial x_k}\delta_{ij} \; , \tag{6}$$

where $\dot{\varepsilon}_{ij}$ is the deviatoric strain rate and $\delta_{ij}$ is the Kronecker delta. All subsequent references to stress or strain rate refer to the deviatoric parts of the two tensors.

## 2.2  Rheology

We use Maxwell visco-elasticity (Maxwell, 1867), where the strain rate is the sum of its elastic and viscous components:

$$\dot{\varepsilon}_{ij} = \dot{\varepsilon}_{ij}^{\mathrm{el}} + \dot{\varepsilon}_{ij}^{\mathrm{vi}} = \frac{1}{2G}\frac{\partial \tau_{ij}}{\partial t} + \frac{1}{2\eta}\tau_{ij} \; , \tag{7}$$

where $\dot{\varepsilon}_{ij}^{\mathrm{el}}$ is the elastic strain rate component, $G$ is the shear modulus, and $\eta$ is the effective shear viscosity. Whereas the elastic deformation is governed by the shear modulus, viscous deformation is a combination of diffusion creep, dislocation creep, and

low-temperature plasticity. Following the approach of Maxwell, we consider all viscous mechanisms in series, which implies that deformation is dominated by the weakest one and that strain rate components are added (Jóźwiak et al., 2015):

$$\dot{\varepsilon}_{\mathrm{II}}^{\mathrm{vi}} = \dot{\varepsilon}_{\mathrm{II}}^{\mathrm{dif}} + \dot{\varepsilon}_{\mathrm{II}}^{\mathrm{dis}} + \dot{\varepsilon}_{\mathrm{II}}^{\mathrm{LTP}} \ , \tag{8}$$

where the superscripts dif, dis and LTP denote diffusion creep, dislocation creep, and low-temperature plasticity, respectively. The subscript II denotes the square root of the second invariant of an arbitrary second-order tensor $C$:

$$C_{\mathrm{II}} = \sqrt{\frac{1}{2} C_{ij} C_{ij}} \ . \tag{9}$$

As a consequence of the Maxwell approach in Eq. (8), the effective viscosity $\eta$ can be expressed as:

$$\eta = \left( \frac{1}{\eta_{\mathrm{dif}}} + \frac{1}{\eta_{\mathrm{dis}}} + \frac{1}{\eta_{\mathrm{LTP}}} \right)^{-1} \ , \tag{10}$$

where

$$\eta_{\mathrm{dif}} = \frac{1}{2} (A_{\mathrm{dif}})^{-1} d^m \exp \left( \frac{E_{\mathrm{dif}}}{RT} \right) \ , \tag{11}$$

$$\eta_{\mathrm{dis}} = \frac{1}{2} (A_{\mathrm{dis}})^{-\frac{1}{n}} (\dot{\varepsilon}_{\mathrm{II}}^{\mathrm{dis}})^{\frac{1}{n}-1} \exp \left( \frac{E_{\mathrm{dis}}}{nRT} \right) \ , \tag{12}$$

$$\eta_{\mathrm{LTP}} = \frac{\tau_{\mathrm{LTP}}}{2 \dot{\varepsilon}_{\mathrm{II}}^{\mathrm{LTP}}} \ . \tag{13}$$

$A$ is a prefactor, $E$ is the activation enthalpy, $d$ is grain size, $m$ is the grain size exponent of diffusion creep, $R$ is the universal
gas constant, and $n$ is the powerlaw exponent of dislocation creep. The LTP-stress $\tau_{\mathrm{LTP}}$ is given by:

$$\tau_{\mathrm{LTP}} = \frac{RT}{E_{\mathrm{LTP}}} \sigma_{\mathrm{res}} \sinh^{-1} \left[ \frac{\dot{\varepsilon}_{\mathrm{II}}^{\mathrm{LTP}}}{A_{\mathrm{LTP}}} \exp \left( \frac{E_{\mathrm{LTP}}}{RT} \right) \right] + \sigma_{\mathrm{b}} \ , \tag{14}$$

$$\sigma_{\mathrm{res}} = \sigma_{\mathrm{L}} + \frac{\sigma_{\mathrm{K}}}{\sqrt{d}} \ , \tag{15}$$

where $\sigma_{\mathrm{b}}$, $\sigma_{\mathrm{L}}$ and $\sigma_{\mathrm{K}}$ are material constants (Hansen et al., 2019). Diffusion creep dominates deformation at low stress/high temperature and dislocation creep at medium stress/temperature. LTP is dominant at large stress/low temperature, and it behaves
similarly to perfect plasticity due to brittle deformation. It remains inactive below a critical stress, but accommodates all deformation that would otherwise increase stress beyond this threshold.

## 2.3 Model setup

We use models with simple shear boundary conditions and central weak inclusions to initialize localization of deformation. Heat flux is zero across all domain boundaries. In 1D, the weak zone is introduced by multiplying the flow law prefactors $A_{\mathrm{dif}}$

and $A_{\mathrm{dis}}$ (see Sect. 2.2) by a weakening factor $\omega$ which follows a Gaussian distribution with a minimum of 1 and a maximum of 2 (Fig. 2a). The full-width-half-maximum of the distribution is 200 m, and the extent of the entire model is 10 km.

The vertical and horizontal extents of the 2D model are 10 km and 60 km, respectively. The weak inclusion is an ellipse with semi-major axes of 375 m and 125 m, respectively. Within this anomaly, $A_{\mathrm{dif}}$ and $A_{\mathrm{dis}}$ are multiplied by 2, and $\sigma_{\mathrm{b}}$ is divided by 2. The different implementations of the weak inclusion are discussed in Sect. 5.2. The lateral boundary conditions in the 2D

model are periodic (Fig. 2b). Unless stated otherwise, the material parameters used in all models are listed in Table 1.

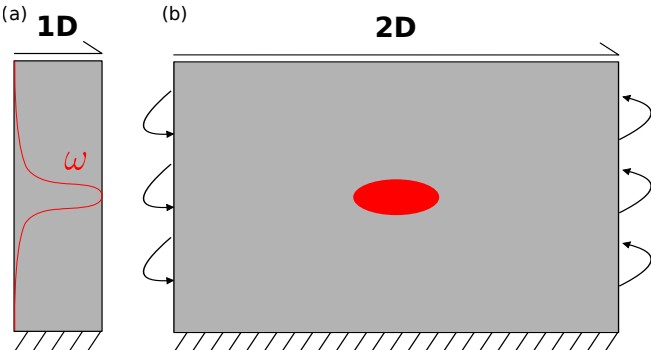

**Figure 2.** Model setups with simple shear boundary conditions. (a) 1D model. Note that the model only has a single cell in the horizontal direction. The red line indicates the distribution of the weakening factor $\omega$. (b) 2D model. The red ellipse indicates the weak inclusion where the weakening factor $\omega$ is applied. Lateral boundaries are periodic. Both setups are not drawn to scale. Vertical extent is 10 km for both models and horizontal extent is 60 km for the 2D model. Adapted from Spang et al. (2024) and Spang et al. (2025a).

### 2.4    The 1D case

In the 1D configuration, the spatial dimensions are reduced to the vertical y-direction, and equations (1) - (6) are simplified. With simple shear boundary conditions, no gravity, and no thermal expansion, the divergence of velocity is inherently zero. The conservation of mass (Eq. (2)) simplifies to:

$$\frac{\partial \rho}{\partial t} = 0 \,, \tag{16}$$

and Eq. (4) simplifies to:

$$\frac{\partial P}{\partial t} = 0. \tag{17}$$

This renders the model incompressible, with density and pressure constant in time. Furthermore, the velocity vector is reduced to its horizontal component, which simplifies Eq. (6) to:

**Table 1.** Material parameters for the reference model. Bracketed superscripts denote the sources of the parameters which are given at the bottom of the table.

| Parameter | Unit | Value | Explanation |
|---|---|---|---|
| $T_0$ | $[^\circ C]$ | 600 | Background temperature |
| $P_0$ | [GPa] | 10 | Background pressure |
| $\dot{\varepsilon}_{\mathrm{bg}}$ | $[\mathrm{s}^{-1}]$ | $5 \times 10^{-13}$ | Background strain rate |
| $\rho_0$ | $[\mathrm{kg\,m}^{-3}]$ | 3300 | Reference density |
| $d$ | $[\mu\mathrm{m}]$ | 100 | Grain size |
| $\eta_{\mathrm{reg}}$ | $[\mathrm{Pa\cdot s}]$ | $10^{15}$ | Regularization viscosity |
| $G$ | [GPa] | 80 | Shear modulus |
| $K_{\mathrm{b}}$ | [GPa] | $133.\overline{3}$ [4] | Bulk modulus |
| $m$ | | 3 [1] | Grain size exponent |
| $A_{\mathrm{dif}}$ | $[\mu\mathrm{m}^m\,\mathrm{MPa}^{-1}\,\mathrm{s}^{-1}]$ | $1.5 \times 10^9$ [1] | Prefactor |
| $E_{\mathrm{dif}}$ | $[\mathrm{kJ\,mol}^{-1}]$ | 375 [1] | Activation enthalpy |
| $n$ | | 3.5 [1] | Stress exponent |
| $A_{\mathrm{dis}}$ | $[\mathrm{MPa}^{-n}\,\mathrm{s}^{-1}]$ | $1.1 \times 10^5$ [1] | Prefactor |
| $E_{\mathrm{dis}}$ | $[\mathrm{kJ\,mol}^{-1}]$ | 530 [1] | Activation energy |
| $A_{\mathrm{LTP}}$ | $[\mathrm{s}^{-1}]$ | $5 \times 10^{20}$ [2] | Prefactor |
| $E_{\mathrm{LTP}}$ | $[\mathrm{kJ\,mol}^{-1}]$ | 550 [2] | Activation energy |
| $\sigma_{\mathrm{L}}$ | [GPa] | 3.1 [2] | Lattice friction |
| $\sigma_{\mathrm{K}}$ | $[\mathrm{GPa}\,\mu\mathrm{m}^{0.5}]$ | 3.2 [2] | Material constant |
| $\sigma_{\mathrm{b}}$ | [GPa] | 1.8 [2] | Back stress |
| $C_p$ | $[\mathrm{J\,kg}^{-1}\,\mathrm{K}^{-1}]$ | 1000 | Heat capacity |
| $k$ | $[\mathrm{J\,s}^{-1}\,\mathrm{m}^{-1}\,\mathrm{K}^{-1}]$ | 3 | Thermal conductivity |
| $H_{\mathrm{L}}$ | $[\mathrm{kJ\,kg}^{-1}]$ | 300 [3] | Latent heat |

[1] Hirth and Kohlstedt (2003), [2] Hansen et al. (2019), [3] Schmeling et al. (2019).

[4] Computed from $G$ and $\nu = 0.25$.

$$\dot{\varepsilon}_{\mathrm{xy}} = \frac{1}{2}\frac{\partial v_x}{\partial y} \,, \tag{18}$$

with the other components of the strain rate and stress tensor equal to zero. This simplifies the conservation of momentum (Eq. (1)) to:

$$\frac{\partial \tau_{\mathrm{xy}}}{\partial y} = 0 \,. \tag{19}$$

## 3 Implementation

The governing equations are discretized on a staggered grid (e.g., Gerya and Yuen, 2003) using the small strain approximation. They are solved with a conservative finite-difference scheme in an iterative manner using the APT method (Frankel, 1950; Räss et al., 2022; Alkhimenkov and Podladchikov, 2024). The code is implemented in the Julia programming language and employs the GEOPARAMS.JL package (Kaus et al., 2023) for parameter nondimensionalization. The 2D implementation further leverages the PARALLELSTENCIL.JL package (Omlin and Räss, 2024) to automatically generate parallel kernels on both central

processing unit (CPU) and graphics processing unit (GPU) devices.

### 3.1 Spatial discretization

For both the 1D and 2D models, we employ a variable grid, with the smallest vertical cell size in the center of the domain. In the 1D models, the central quarter of the grid consists of uniformly sized cells, while spacing increases linearly towards the model boundaries. The outermost cells are approximately 125 times larger than those at the center, allowing for maximum resolution

in the region where thermal runaway is expected to occur. This is common practice when investigating thermal runaway (e.g., Thielmann et al., 2015). Material properties and most field variables are defined at cell centers, whereas velocity and heat flux are located at cell edges (Fig. A2a).

In 2D, the grid refinement is limited to a factor of 2 to avoid convergence issues arising from large cell aspect ratios. In the horizontal direction, all cells are the same size. We use 1536 and 256 cells in the horizontal and vertical direction respectively,

yielding resolutions of about 39 m (horizontal) and 26 - 52 m (vertical). We use a staggered grid approach where material properties, temperature, pressure, viscous dissipation, and normal stress components are defined at cell centers. Velocity and heat flux are defined on cell edges, while shear stress components are located at cell corners (Fig. A2b).

### 3.2 Accelerated pseudo-transient method

In the APT approach, the conservation equations are solved at each physical time step by introducing a pseudo-time derivative

for each equation and iteratively updating the primary variables $v$, $P$, and $T$ until the residuals drop below a given numerical tolerance. Applying this procedure to Eq. (1)-(3) yields:

$$\frac{\partial v_i}{\partial \psi} = \frac{\partial \tau_{ij}}{\partial x_j} - \frac{\partial P}{\partial x_i} \, , \tag{20}$$

$$\frac{\partial P}{\partial \psi} = \frac{1}{K_b} \frac{\partial P}{\partial t} + \frac{\partial v_i}{\partial x_i} \, , \tag{21}$$

$$\frac{\partial T}{\partial \psi} = \frac{1}{\rho C_p} \left[ \frac{\partial}{\partial x_i} \left( k \frac{\partial T}{\partial x_i} \right) + \tau_{ij} \dot{\varepsilon}_{ij}^{vi} \right] - \frac{\partial T}{\partial t} \, , \tag{22}$$

where $\partial / \partial \psi$ denotes the pseudo-time derivative. During each pseudo-time iteration, each primary variable is incremented proportionally to the sum of the current residual and the previous increment (Duretz et al., 2019):

$$\Delta_\gamma = \left[ \frac{\partial \gamma}{\partial \psi} + \left( 1 - \frac{1}{\zeta_\gamma} \right) \Delta_\gamma^{\text{prev}} \right] \Delta \psi_\gamma \, , \tag{23}$$

where $\gamma$ represents one of the primary variables $v$, $P$, or $T$, $\Delta_\gamma$ is the current increment of the respective variable, $\Delta_\gamma^{\text{prev}}$ is the increment from the previous iteration, $\zeta_\gamma$ is the damping parameter ($> 1$). $\Delta \psi_\gamma$ is the size of the pseudo-time step given by:

$$\Delta \psi_{v_i} = \frac{\Delta x_i}{f_v \eta} \, , \tag{24}$$

$$\Delta \psi_P = \frac{f_P \eta}{\max(\text{nc}_i)} \, , \tag{25}$$

$$\Delta \psi_T = \min \left( \frac{\min(\Delta x_i)^2}{2 \, n_{\text{dim}} \kappa}, \frac{\Delta t}{2} \right) \, , \tag{26}$$

where $\Delta x_i$ is the grid spacing, $f_v$ and $f_P$ are factors, $\text{nc}_i$ is the number of cells in each dimension, $n_{\text{dim}}$ is the number of dimensions, $\kappa = k/(\rho C_\text{p})$ and $\Delta t$ is the physical time step.

The left hand side terms in Eq. (20) - (22) are equivalent to the residuals of the conservation equations. Once all of them are smaller than a given numerical tolerance of $10^{-6}$ after normalization, the solution is converged and is equivalent to a fully implicit, backward Euler solution with converged nonlinearities.

### 3.3 Viscosity update

Given the nonlinear nature of dislocation creep and low-temperature plasticity, the strain rate partitioning (Eq. (8)) cannot be
solved analytically but requires a numerical approach. It can be updated and solved alongside the conservation equations (20) - (22). To stabilize the rheology solver, we use a relaxation approach for the viscosity updates of each mechanism during the pseudo-transient (PT) iterations:

$$\eta_i^{\text{it}} = \exp \left[ (1 - \eta_{\text{rel}}) \log(\eta_i^{\text{it}-1}) + \eta_{\text{rel}} \log(\eta_i^{\text{t}}) \right] \, , \tag{27}$$

where the superscript it denotes the iteration count, $\eta_{\text{rel}} < 1$ is the relaxation factor (Duretz et al., 2019), and $\eta_i^{\text{t}}$ is the target
viscosity (i.e. the new viscosity without relaxation). We discuss our strategy for solving the strain rate partitioning in Appendix A and Fig. A1.

### 3.4 Regularization

To stabilize the model during thermal runaway and mitigate mesh dependency, we test three regularization strategies: (i) viscosity regularization, (ii) gradient regularization, and (iii) inclusion of latent heat of melting. All approaches aim to limit
maximum strain rates and prevent viscosities from dropping below a critical threshold. We note that alternative regularization strategies for brittle failure have been proposed in the literature (e.g., Duretz et al., 2023; Goudarzi et al., 2023; Gerolymatou et al., 2024). We discuss the strategies' relation to physical mechanisms in Sect. 4.3.5.

### 3.4.1 Viscosity regularization

Viscosity regularization imposes a direct lower bound on viscosity, effectively stopping the self-softening behavior of thermal runaway once this threshold is reached. To implement this, we modify equation (10) as follows:

$$\eta = \left( \frac{1}{\eta_{\mathrm{dif}}} + \frac{1}{\eta_{\mathrm{dis}}} + \frac{1}{\eta_{\mathrm{LTP}}} \right)^{-1} + \eta_{\mathrm{reg}} , \tag{28}$$

where $\eta_{\mathrm{reg}}$ is the regularization viscosity. This approach has been previously applied to regularize brittle plasticity (Duretz et al., 2020; Jacquey and Cacace, 2020; Kiss et al., 2023; Alkhimenkov et al., 2024) and rate-and-state friction models (Pranger et al., 2022; Goudarzi et al., 2023). Our rheological model is illustrated in Fig. A1.

### 3.4.2 Gradient regularization

In gradient regularization, the viscous dissipation is distributed over a broader area, which limits localized temperature increase, viscosity reduction, and strain localization. This is achieved by introducing a diffusion term to the shear heating component of the conservation of energy (Eq. (3)):

$$\rho C_p \frac{\partial T}{\partial t} = \frac{\partial}{\partial x_i} \left( k \frac{\partial T}{\partial x_i} \right) + \tau_{ij} \left( \dot{\varepsilon}_{ij}^{\mathrm{vi}} + \lambda_{\mathrm{reg}}^2 \frac{\partial^2 \dot{\varepsilon}_{ij}^{\mathrm{vi}}}{\partial x_i^2} \right) , \tag{29}$$

where $\lambda_{\mathrm{reg}}$ is a regularizing diffusion length scale. With increasing $\lambda_{\mathrm{reg}}$, the dissipation is smoothed over a larger area and thermal runaway will be damped. This approach has also been employed in the regularization of rate-and-state friction models (Sleep, 1997; Pranger et al., 2022) and tested in the context of brittle faulting (De Borst and Mühlhaus, 1992; Duretz et al., 2023).

### 3.4.3 Inclusion of latent heat of melting

Melting is an endothermic process and as such, it can act as an energy sink at large temperatures. This could potentially offset the viscous dissipation term and limit temperature growth, consequently stopping the self-softening behavior of thermal runaway like viscosity regularization. To introduce this process into the governing equations, we add a term to Eq. 3 to account for the energy consumed by melting:

$$\rho C_p \frac{\partial T}{\partial t} = \frac{\partial}{\partial x_i} \left( k \frac{\partial T}{\partial x_i} \right) + \tau_{ij} \dot{\varepsilon}_{ij}^{\mathrm{vi}} - \rho H_{\mathrm{L}} \frac{\partial F}{\partial t} , \tag{30}$$

where $H_{\mathrm{L}}$ is latent heat and $F$ is the melt fraction (Schmeling et al., 2019). Melt fraction is a function of pressure and temperature and is computed after a parameterization for anhydrous melting of peridotite (Katz et al., 2003). Similarly to the viscosity, the melt fraction has to be updated incrementally during the PT iterations:

$$F^{\text{it}} = (1 - F_{\text{rel}})F^{\text{it}-1} + F_{\text{rel}}F^{\text{t}} \, , \tag{31}$$

where $F^{\text{it}}$ and $F^{\text{it}-1}$ are the melt fraction in the current and previous iteration, respectively, $F^{\text{t}}$ is the target melt fraction according to the melting model, and $F_{\text{rel}} = 10^{-4}$ is a relaxation factor.

A complete description of melting would also involve changes to the conservation of mass as well as a feedback on rheology. As we are interested in the potential of melting as a regularization, we neglect these components, since the weakening effect of partial melt on rheology would increase runaway intensity.

## 4 Numerical challenges and solution strategies

We use 1D models to illustrate the numerical challenges associated with rapid strain localization and the strategies we employ to address them. Similar problems arise in 2D models, which are discussed in Sect. 5. The primary challenges include: (i) selecting appropriate time steps to accurately capture the runaway phase; (ii) avoiding round-off errors caused by abrupt shifts in the model's characteristic time scales; (iii) maintaining solver stability during runaway; and (iv) minimizing resolution dependence.

### 4.1 Adaptive time stepping

The basic model behavior is described in Fig. 1b and the introduction. Outside the runaway phase, time steps ranging from tens to thousands of years are sufficient. However, resolving the thermal runaway phase requires time steps on the order of milliseconds. While large time steps may adequately capture the long-term stress evolution, they fail to resolve the transient dynamics leading up to runaway (Fig. 3). In particular, they significantly underestimate temperature increase and slip velocity (Fig. 3b,d).

As the spontaneous onset of runaway cannot be predicted a priori, an adaptive time-stepping scheme is critical. Identifying suitable time steps is a well-known challenge in scientific computing, and a number of studies propose different methods (e.g., Bursi and Shing, 1996; Rylander and Bondeson, 2002; Ropp et al., 2004; Söderlind and Wang, 2006). As thermal runaway is driven by the conversion of elastic energy to thermal energy (e.g. Ogawa, 1987; Spang et al., 2024), the most indicative parameters for the onset and intensity of runaway are stress and temperature. Therefore, we constrain time steps by limiting the maximum allowable change in these two quantities ($\Delta\tau$, $\Delta T$). Similar strategies are employed in earthquake modeling studies (Herrendörfer et al., 2018; Dal Zilio et al., 2022; Pranger et al., 2022). We evaluate three methods for implementing adaptive time stepping. In all cases, we define threshold values of $\Delta\tau_{\text{max}} = 50\,\text{MPa}$ and $\Delta T_{\text{max}} = 5\,\text{K}$.

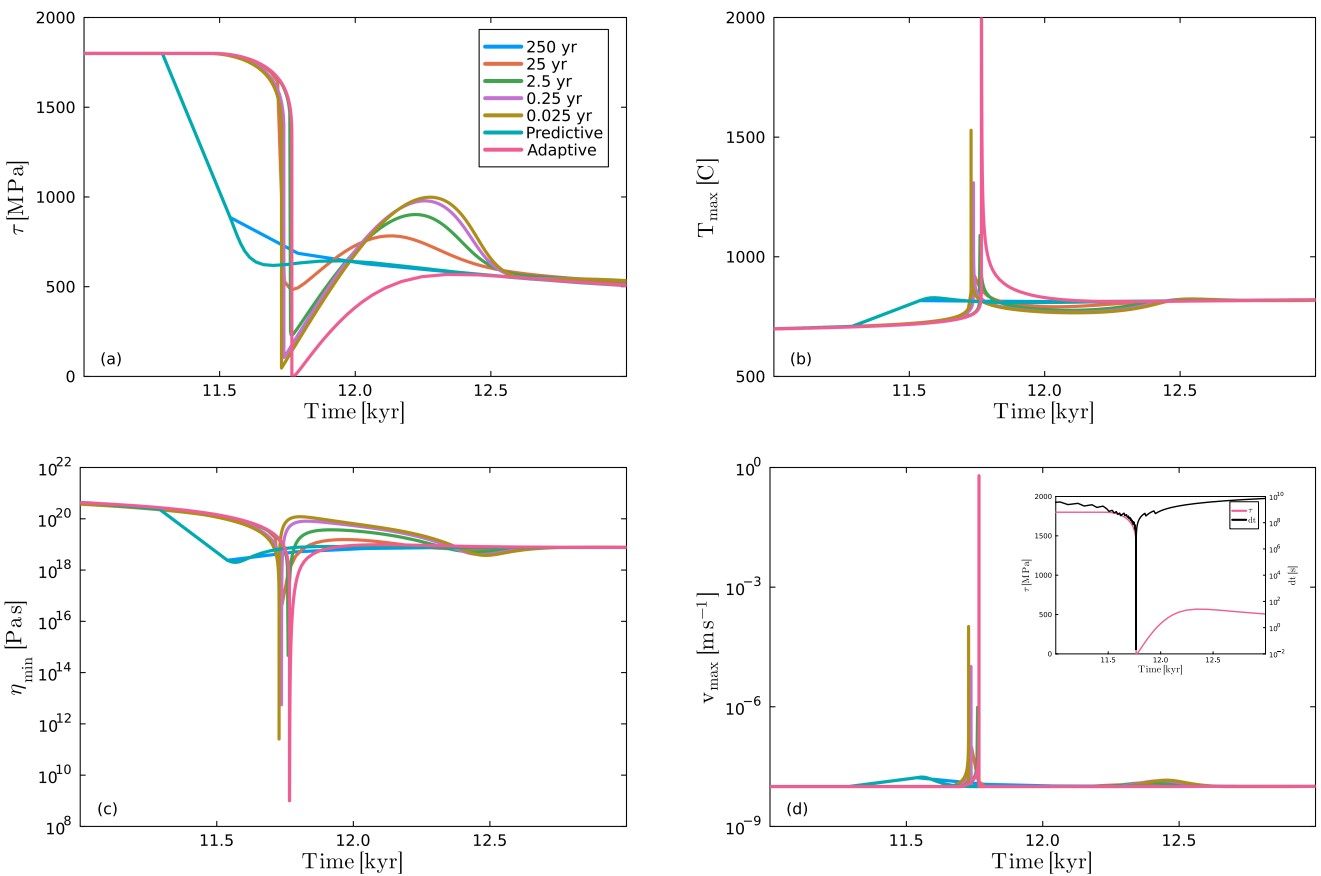

**Figure 3.** Model results for different time stepping schemes. (a) Temporal stress evolution as a function of fixed time steps of different size in comparison to linear-predictive (Sect. 4.1.1) and restarting-adaptive (Sect. 4.1.3) time stepping. (b) Maximum temperature. (c) Minimum viscosity. (d) Maximum velocity. Inset shows temporal evolution of stress and time step, using the restarting-adaptive method. $\eta_{\mathrm{reg}} = 10^9$ Pa·s.

### 4.1.1 Linear-predictive

For the linear-predictive scheme, we assume that the rates of change of stress $\tau$ and temperature $T$ do not increase significantly in the subsequent time step. Based on this assumption, the new time step can be determined by scaling the previous time step according to the changes in $\tau$ and $T$:

$$\Delta t^{\mathrm{nt}} = \Delta t^{\mathrm{nt}-1} \min\left( \frac{\Delta T_{\max}}{\Delta T^{\mathrm{nt}-1}}, \frac{\Delta \tau_{\max}}{\Delta \tau^{\mathrm{nt}-1}}, 1.25 \right) , \tag{32}$$

where $\Delta t^{\mathrm{nt}}$ is the upcoming time step, $\Delta t^{\mathrm{nt}-1}$ is the previous time step, $\Delta T^{\mathrm{nt}-1}$ is the maximum temperature change during 250 the previous step, and $\Delta \tau^{\mathrm{nt}-1}$ is the stress change (which is spatially uniform in the 1D domain). To avoid excessive time

step increases, we added the factor of 1.25 in Eq. (32). It limits the time step growth to 25% of the previous value when $\Delta t$ is predicted to increase.

If the actual rates of change in $\tau$ and $T$ increase, the resulting $\Delta \tau$ and/or $\Delta T$ may exceed their respective thresholds $\Delta \tau_{\mathrm{max}}$ and $\Delta T_{\mathrm{max}}$. This causes the subsequent time step to be shorter. However, due to the highly nonlinear nature of thermal runaway, this predictive scheme may be inadequate, failing to decrease the time step fast enough once localization and stress release begin (Fig. 3a).

### 4.1.2 Iteration-adaptive

In the iteration-adaptive scheme, we also use Eq. (32) to predict the new time step. However, instead of applying it only once at the beginning of a physical time step, we dynamically adjust the time step during every iteration of the APT solver. This approach enables rapid reduction of the time step by several orders of magnitude within a single physical time step, while still adhering to the constraints set by $\Delta \tau_{\mathrm{max}}$ and $\Delta T_{\mathrm{max}}$. However, as the elastic component of the strain rate is time step-dependent (Eq. (7)), adapting the time step during PT iterations can lead to unstable behavior where the residuals oscillate and fail to converge. As this method is unstable, we did not plot it in Fig. 3a.

### 4.1.3 Restarting-adaptive

In the restarting-adaptive scheme, we rely on Eq. (32) to evaluate the appropriate time step during PT iterations. However, unlike the iteration-adaptive approach, the time step is not reduced within the PT iterations. Instead, if Eq. (32) indicates that the current time step is too large, the entire physical time step is restarted with a reduced (by a factor of 2) step size. To facilitate this, all relevant fields – stress, temperature, pressure, density, viscosity, and velocity – are saved at the start of each new time step. If a restart is triggered, these values are restored, and the time step is recalculated.

Multiple restarts per time step are possible and often necessary during the onset of thermal runaway. This strategy is effective in ensuring solver stability while rapidly adapting the time step. Its primary drawback is that some redundant computations occur during restarts. However, the redundancy is generally small compared to the overall computations (and iterations) required to solve each time step.

The inset in Fig. 3d illustrates the performance of this method. The time step initially remains on the order of hundreds of years during the steady-state creep phase, drops by approximately two orders of magnitude as stress begins to relax, and then decreases by another ten orders of magnitude during the onset of thermal runaway. In the elastic reloading phase, $\Delta t$ quickly recovers to hundreds of years as both stress and temperature evolve more slowly.

## 4.2 Adaptive rescaling

Numerical solvers commonly use internal scaling to center quantities around 1 which minimizes round-off errors due to numerical precision. To do so, a set of scales is created, and all dimensional quantities are divided by an appropriate combination of these scales. As an example, a geodynamic model focused on plate-scale deformation might use a time scale of $t_{\mathrm{sc}} = 10^{12}\,\mathrm{s}$

and a stress scale of $\tau_{sc} = 10^8\,\mathrm{Pa}$ which combine to a viscosity scale of $\eta_{sc} = 10^{20}\,\mathrm{Pa \cdot s}$. This means a time step of 100 years is scaled to $\Delta t_{ND} = \frac{\Delta t}{t_{sc}} \approx 0.003156$, where $\Delta t_{ND}$ is the nondimensional time step used inside the solver.

This becomes problematic once adaptive time stepping reduces the dimensional time step to one second, as this is equivalent to $10^{-12}$ after scaling. Considering the numerical precision of $10^{-15}$, such a low value is prone to round-off errors. To mitigate this, we decrease $t_{sc}$ by one order of magnitude every time the nondimensional time step drops below $10^{-9}$. Changing the time scale does not only impact the nondimensional time step but all quantities which carry units of seconds such as strain rates, velocities, and viscosities. All of these have to be rescaled together. This is convenient for viscosities as they also decrease significantly during runaway. It is also beneficial for velocities and strain rates as they increase during runaway and decreasing

$t_{sc}$ increases the velocity and strain rate scales.

In Fig. 4, we demonstrate how rescaling facilitates convergence and reduces the number of iterations by two orders of magnitude for a model that takes time steps as low as $25\,\mu s$. Without rescaling, the model requires about $2 \times 10^9$ iterations in total to solve, the majority of them during thermal runaway. Rescaling properties with time scales in their units as soon as the nondimensional time step $\Delta t_{ND}$ drops below $10^{-14}$ reduces the number of iterations by one order of magnitude. Rescaling at

295 $\Delta t_{ND} < 10^{-12}$ reduces the total number of iterations by another order of magnitude, and only half of them are used during the runaway. Further reduction of the critical $\Delta t_{ND}$ has only negligible effects (Fig. 4) despite the proximity of the values to numerical precision.

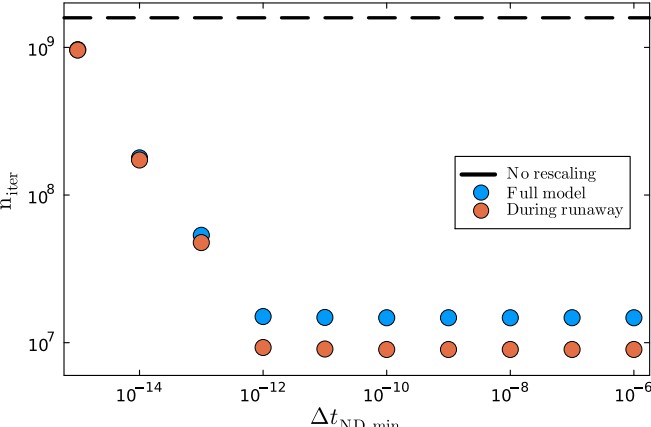

**Figure 4.** Effect of adaptive rescaling. Sum of iterations for full model (blue) and during runaway (orange) as a function of the minimum allowed $\Delta t_{ND}$ before rescaling is used to increase it. The dashed black line shows the number of iterations without any rescaling. Note that the models with $\Delta t_{ND,min} = 10^{-15}$ and no rescaling have many non-converged time steps. All models have identical results in terms of stress, temperature, and velocity. $\eta_{reg} = 10^6$ Pa·s.

## 4.3 Regularization

During thermal runaway, the viscosity within the shear zone decreases dramatically (more than 10 orders of magnitude) due to the temperature increase. Large contrasts in material properties are challenging for numerical solvers (e.g., Gerya, 2019), especially for iterative approaches which rely on local conditioning. Elasticity can reduce the stiffness contrast between high and low viscosity areas, but this is not sufficient to guarantee convergence. Even if the solver converges, shear zones often thin to the width of one grid cell. In this case, the mechanical behavior of the model is governed by the numerical resolution instead of the physics of the problem (De Borst et al., 1993; Iordache and Willam, 1998; Jacquey et al., 2021). To alleviate this issue and improve reproducibility, we test three regularization methods: viscosity regularization (see Sect. 3.4.1), gradient regularization (see Sect. 3.4.2), and latent heat of melting (see Sect. 3.4.3). To quantify the impact of viscosity and gradient regularization, we run 60 1D simulations in which we vary between five different numerical resolutions (63-1023 cells), with six different viscosity regularization values $\eta_{\mathrm{reg}}$ ($10^6$-$10^{18}$ Pa·s), and six different gradient regularization values $\lambda_{\mathrm{reg}}$ (1-32 m). As latent heat proved to be unable to regularize the reference model, we did not include it in this parameter study. We use maximum velocity $v_{\mathrm{max}}$, maximum temperature $T_{\mathrm{max}}$, and shear zone width $d_{\mathrm{sz}}$ as diagnostic parameters for the analysis.

### 4.3.1 Viscosity regularization

Applying viscosity regularization renders the diagnostic parameters resolution independent (Fig. 5a,b,c). Instead, these quantities exhibit a strong, exponential dependence on the regularization viscosity $\eta_{\mathrm{reg}}$. For $\eta_{\mathrm{reg}} \geq 10^{12}$ Pa·s, these quantities remain nearly identical across all tested grid resolutions, ranging from 63 to 1023 cells (corresponding to minimum cell sizes between 2 m and 0.125 m). At $\eta_{\mathrm{reg}} = 10^{12}$ Pa·s, the shear zone localizes to a single grid cell in the coarsest model (63 cells; blue curve in Fig. 5). For lower values of $\eta_{\mathrm{reg}}$, results from this low-resolution model begin to diverge from those of finer grids. As $\eta_{\mathrm{reg}}$ is further reduced, finer-resolved models also localize to a single cell and their results start to diverge from models that can still resolve the shear zone.

Once a model localizes deformation to a single grid cell, both $d_{\mathrm{sz}}$ and $T_{\mathrm{max}}$ plateau and cease to vary with decreasing $\eta_{\mathrm{reg}}$ (Fig. 5b,c). In contrast, $v_{\mathrm{max}}$ continues to increase as $\eta_{\mathrm{reg}}$ decreases, but it also slowly diverges from models that are still resolved.

The total number of PT iterations $\mathrm{n_{iter}}$, normalized by grid resolution, decreases with increasing $\eta_{\mathrm{reg}}$, reflecting the fact that a more strongly regularized runaway is numerically easier to solve (Fig. 5d). Higher-resolution models exhibit slightly more efficient convergence compared to lower-resolution counterparts.

The temporal evolution of stress remains largely unaffected by variations in $\eta_{\mathrm{reg}}$. For $\eta_{\mathrm{reg}} \leq 10^{15}$ Pa·s, the models consistently exhibit rapid and complete stress relaxation. In contrast, $\eta_{\mathrm{reg}} = 10^{18}$ Pa·s leads to slower and incomplete relaxation (inset in Fig. 5a). This trend is observed across all resolutions. Similar effects of viscosity regularization have been reported by Spang et al. (2024).

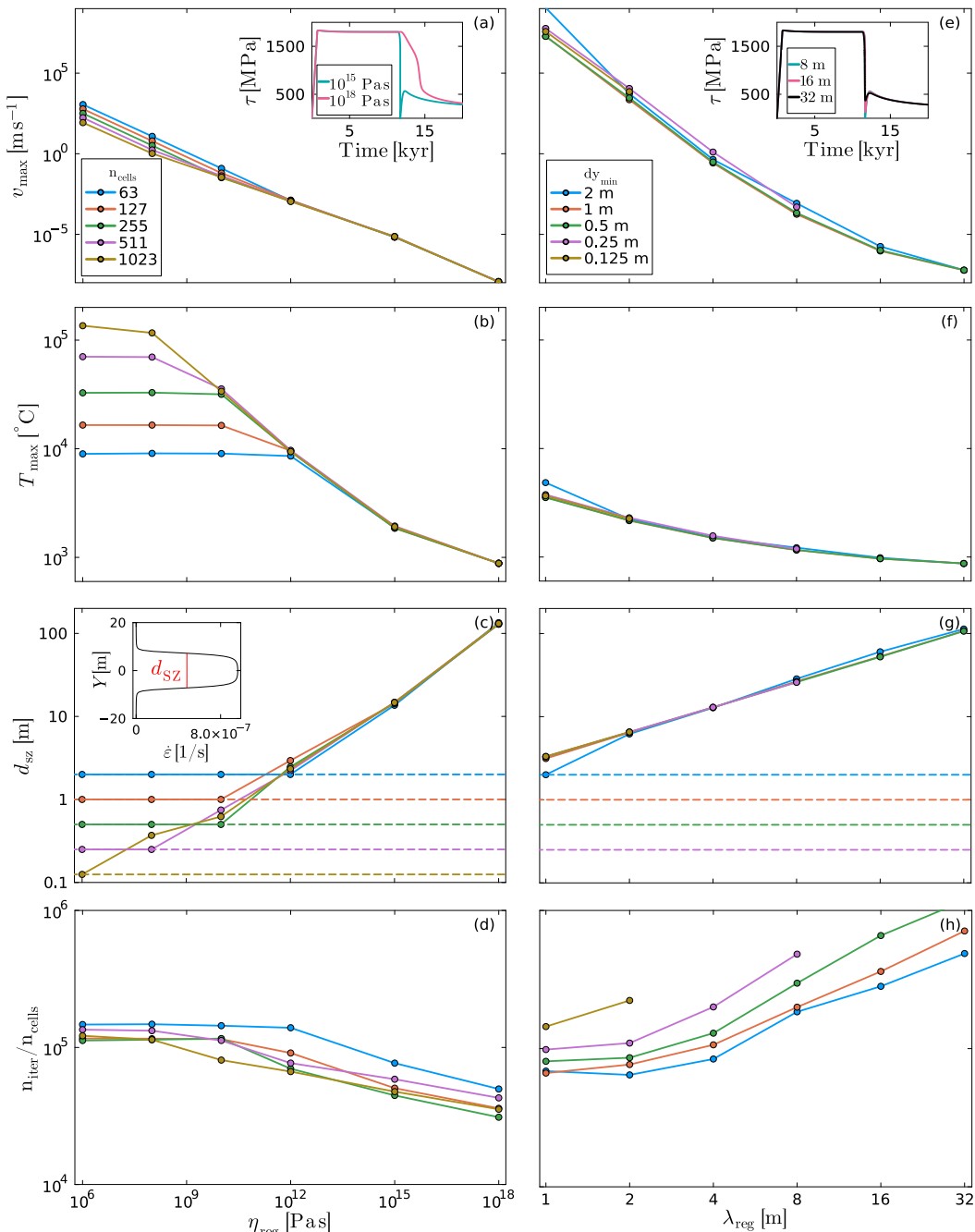

**Figure 5.** Effect of viscosity regularization (left column) and gradient regularization (right column). Colors correspond to resolution (number of cells and corresponding size of smallest cell) and all axes are logarithmic. (a,e) Maximum velocity. (b,f) Maximum temperature. (c,g) Shear zone width (full-width-half-maximum of strain rate peak, see inset). Dashed lines indicate size of one cell for each resolution. (d,h) Total number of iterations divided by number of cells. Insets in (a) and (e) show stress evolution for 255 cells and largest $\eta_{\mathrm{reg}}/\lambda_{\mathrm{reg}}$. All models with lower values are indistinguishable from the displayed ones.

### 4.3.2 Gradient regularization

As in the viscosity regularization case, applying gradient regularization renders the diagnostic parameters resolution independent (Fig. 5e,f,g). Instead, these quantities exhibit a strong, exponential dependence on the regularization diffusion length scale $\lambda_{\mathrm{reg}}$. While minor discrepancies persist between different resolutions, they are negligible compared to the variations induced by changes in $\lambda_{\mathrm{reg}}$. One exception is the coarsest model (63 grid cells) with $\lambda_{\mathrm{reg}} = 1$ m, which slightly overestimates both $v_{\mathrm{max}}$ and $T_{\mathrm{max}}$. In this case, the shear zone has localized to a single grid cell (Fig. 5g).

Across the tested range of $\lambda_{\mathrm{reg}}$ ($1 - 32$ m), $v_{\mathrm{max}}$ spans from $10^{-7}$ and $10^{7}\,\mathrm{m\,s^{-1}}$, $T_{\mathrm{max}}$ ranges between 800 and 4000 $^\circ$C, and $d_{\mathrm{sz}}$ varies from approximately 3 to 100 m. Somewhat counterintuitively, larger values of $\lambda_{\mathrm{reg}}$ – which prevent extreme localization resulting in a more attenuated runaway – require more PT iterations resulting in larger solution time (Fig. 5h). Moreover, the number of iterations per grid cell increases with numerical resolution. Models with 511 grid cells and $\lambda_{\mathrm{reg}} > 8$ m, as well as 1023-cell models with $\lambda_{\mathrm{reg}} > 2$ m did not complete in one day and are not shown in Fig. 5. We discuss the reasons for this in Sect. 4.3.4.

For $\lambda_{\mathrm{reg}} \leq 8$ m, stresses relax rapidly and nearly completely. In contrast, for $\lambda_{\mathrm{reg}} > 8$ m, residual stresses of several hundred MPa remain at the end of the thermal runaway phase (inset in Fig. 5e).

### 4.3.3 Inclusion of latent heat of melting

To test the potential of melting as a regularization, we repeat the reference model (Fig. 1b) with the changes described in Sect. 3.4.3 and without the previously mentioned regularization methods. Once the shear zone reaches the solidus of about 1900 $^\circ$C (at $P_0 = 10$ GPa), temperature increase slows down as thermal energy partitions into melting (Fig. 6a). After about one millisecond, the shear zone is completely molten and temperature continues to increase with the same rate as below the solidus since no additional energy can be partitioned into melting. Overall, the inclusion of latent heat has no significant impact on the model evolution. Results are similar in our 2D models (Fig. 6b).

### 4.3.4 Comparison

Viscosity and gradient regularization achieve the same overarching goal: they effectively attenuate thermal runaway, ensure numerical stability, and provide control over the degree of strain localization. By doing so, they eliminate the dependence of diagnostic parameters on spatial resolution, making quantities such as $v_{\mathrm{max}}$, $T_{\mathrm{max}}$, and $d_{\mathrm{sz}}$ primarily functions of $\eta_{\mathrm{reg}}$ or $\lambda_{\mathrm{reg}}$ instead. This control breaks down when the shear zone narrows to a single grid cell. At that point, regularization can no longer constrain the degree of localization, and resolution-dependent artifacts reappear.

A direct, quantitative comparison between the two methods is not straightforward, as there is no known correspondence between specific values of $\eta_{\mathrm{reg}}$ and $\lambda_{\mathrm{reg}}$. Nevertheless, a qualitative comparison of the columns two columns in Fig. 5 reveals distinct differences. Gradient regularization allows significantly larger $v_{\mathrm{max}}$ – spanning orders of magnitude beyond values observed with viscosity regularization. However, $T_{\mathrm{max}}$ is approximately two orders of magnitude lower when gradient regularization is employed. Although both approaches produce similar shear zone widths when considering largest regularization

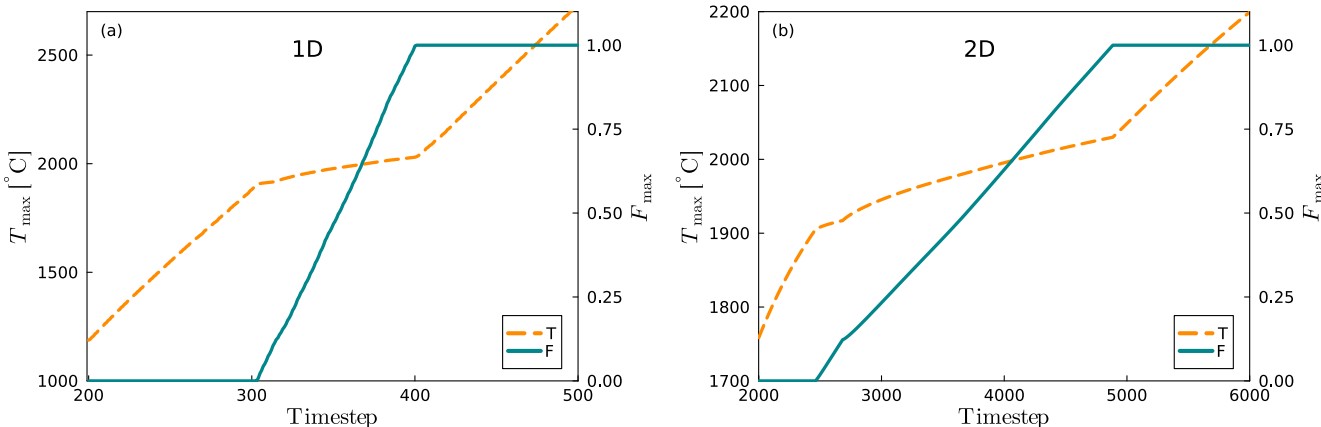

**Figure 6.** Effect of considering the latent heat of melting. Dashed orange line shows evolution of maximum temperature, and turquoise line shows evolution of maximum melt fraction. Note that we plot time step instead of time due to the small size of the time steps ($\sim 1\mu$s) during melting. (a) 1D. (b) 2D.

values, the viscosity-regularized models generate up to an order of magnitude narrower shear zones for the smallest considered regularization values. These differences stem from the fundamentally different ways the two methods constrain localization.

Viscosity regularization allows for the full release of stored elastic energy within the shear zone during stress relaxation, leading to extreme peak temperatures of up to $10^5\,°$C. However, by introducing a lower bound on viscosity, it limits the extent

to which this heating can impact the rheology and weaken the material. As deformation is tightly coupled to rheology, this constraint also limits maximum slip velocities. In contrast, gradient regularization distributes the released energy across a broader region, leading to lower peak temperatures and wider shear zones. Because this method does not impose an explicit lower viscosity bound, extreme deformation rates can still occur.

The computational cost of the two methods also differs significantly, as illustrated by the normalized number of PT itera-

370 tions in Fig. 5d and 5h. At low resolutions and with less pronounced regularization (low $\eta_{\mathrm{reg}}$ and $\lambda_{\mathrm{reg}}$), both methods perform similarly. However, as $\eta_{\mathrm{reg}}$ or $\lambda_{\mathrm{reg}}$ increase, viscosity regularization becomes more efficient, requiring fewer PT iterations. Conversely, gradient regularization becomes increasingly expensive. Larger values of $\lambda_{\mathrm{reg}}$ allow for faster diffusion of dissipative work, effectively reducing the maximum allowed physical time steps.

Resolution scaling further differentiates the two methods. For viscosity regularization, the number of iterations per cell

remains nearly constant with increasing resolution. In contrast, this ratio grows with resolution when gradient regularization is employed, making the latter increasingly impractical for high-resolution simulations. The number of necessary iterations for diffusive processes is known to grow quadratically with the number of cells (e.g., Räss et al., 2022).

Including the latent heat of melting only has a negligible effect on the model evolution as it provides no significant limitation of weakening and localization. A melt fraction of 100% requires about $10^9\,\mathrm{J\,m^{-3}}$ while the shear heating term is about

380 $10^{12}\,\mathrm{J\,m^{-3}\,s^{-1}}$ when the model reaches the solidus. For melting to be effective in attenuating thermal runaway, the melt would

need to be immediately transported out of the shear zone which would remove energy from the shear zone and continuously bring new host rock in contact with the shear zone which can absorb energy by melting as well. This process is indeed observed in pseudotachylytes in the form of injection veins (Rowe et al., 2012; Andersen et al., 2014).

### 4.3.5 Relation to physical mechanisms

Regularization techniques are synthetic additions to physics-based equations. Their intended benefits include numerical stability, reproducibility, mitigation of mesh dependency, smoothening of discontinuities, and simplification of complexity. If they are effective, they provide better control over the model behavior but come with the inherent cost of diverging from the physical solution once they start to affect the model.

Nevertheless, regularization techniques can also be interpreted as a simplification of a physical process that is not part of 390 the model equations. The viscosity regularization could be imagined as a simplification of melting. By acting as a minimum viscosity cut-off, it decouples the deformation in the model from the temperature and the flow laws describing solid-state creep. Melting could have a similar effect, replacing the governing olivine flow laws with a different rheology, potentially temperature- and strain rate-dependent. We note that the values we employ for $\eta_{\mathrm{reg}}$ are significantly larger than the viscosity of peridotite melt (Liebske et al., 2005; Xie et al., 2021).

Gradient regularization distributes the localized shear heating over a larger area. As temperature is mainly governed by shear heating during runaway, this regularization is effectively a smoothing process for temperature. Therefore, it can be compared to an advection scheme which has a similar effect.

Ideally, regularization is replaced by additional physical processes (e.g., melting and melt transport). This requires an accurate description of the physical process by the governing equations, exhaustive experimental constrains on the associated material parameters, and a numerical solver that can handle the additional non-linearity that is potentially introduced. Furthermore, there is no guarantee that additional physical processes are sufficient to regularize a process enough for numerical stability and reliability (Gerolymatou et al., 2024). Additional physical processes that could affect the evolution of our models are grain size evolution and phase transformations. We discuss them in Sect. 6.2 and 6.3.

### 4.4 Viscosity convergence

During the elastic loading phase, the model typically converges within a few ($< 100$) PT iterations. While such fast convergence is computationally efficient, it can introduce numerical errors when using the viscosity relaxation method (Eq. (27)). In this approach, the viscosity is incrementally updated in each iteration using a relaxation factor, commonly $\eta_{\mathrm{rel}} = 0.01$, meaning that only 1% of the computed viscosity update is applied per iteration. Although this under-relaxation stabilizes the solver, it can hinder convergence of the viscosity field for a low PT iteration count.

Figure 7b shows that after 100 iterations, the viscosity update has only progressed about halfway towards its target value. Converging viscosity relaxation (i.e., reaching the updated steady-state value) typically requires around 500 iterations for $\eta_{\mathrm{rel}} = 0.01$. Failing to accurately resolve viscosity relaxation may become problematic near the onset of LTP creep, where $\eta_{\mathrm{LTP}}$ drops rapidly as stress approaches the yield threshold $\tau_{\mathrm{LTP}}$.

LTP accommodates all deformation that would otherwise increase stress beyond this threshold. If $\eta_{\mathrm{LTP}}$ and the associated strain rate partitioning are not updated fast enough, stresses can significantly exceed $\tau_{\mathrm{LTP}}$, requiring corrective adjustments in subsequent time steps (Fig. 7a). This not only leads to an incorrect stress evolution, but can also trigger spurious slip events that would not occur under properly updated stress conditions.

To mitigate this issue, we monitor the convergence between viscosity $\eta^{\mathrm{it}}$ and target viscosity $\eta^{\mathrm{t}}$ (Eq. 27). When the relative difference $\frac{|\eta^{\mathrm{it}}-\eta_{\mathrm{t}}|}{\eta_{\mathrm{t}}}$ is smaller than the viscosity tolerance $\mathrm{tol}_\eta$, viscosity is considered converged. Once the conservation equations (Sect. 3.2), the viscosity, and the strain rate partitioning (Appendix A) are converged, we accept the solution. This ensures that both rheological and mechanical responses are correctly captured during the elastic-to-LTP transition (Fig. 7a).

The stress overshoot for insufficient viscosity convergence is more prominent when the steady-state stresses of diffusion and dislocation creep are large. For the model in Fig. 7a, we increased $E_{\mathrm{dif}}$ and $E_{\mathrm{dis}}$ to 435 and 670 kJ mol$^{-1}$, respectively, which is equivalent to considering the pressure dependence of the rheology (Hirth and Kohlstedt, 2003) and 10 GPa of background pressure (Table 1).

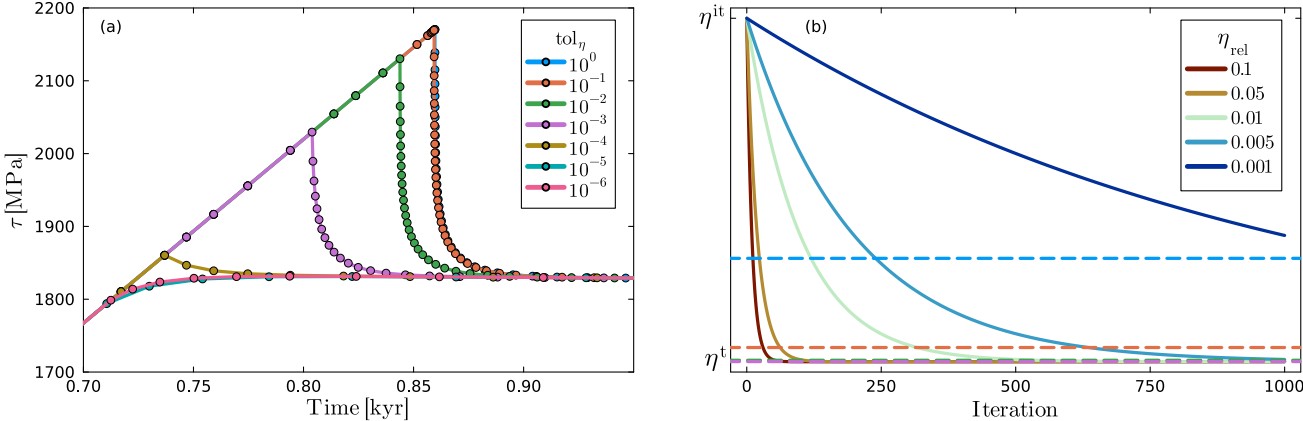

**Figure 7.** Effects of viscosity relaxation. (a) Zoom on transition from elastic loading to LTP in temporal evolution of stress ($\tau$) for different viscosity tolerances ($\mathrm{tol}_\eta$). The stress peak disappears if $\mathrm{tol}_\eta$ is reduced to $\sim 10^{-5}$. (b) Convergence of $\eta^{\mathrm{it}}$ towards $\eta^{\mathrm{t}}$ during the PT iterations according to Eq. 27 for different $\eta_{\mathrm{rel}}$. Dashed lines correspond to the tolerances in (a). As all low-tolerance lines overlap, we did not display $10^{-6}$ to $10^{-4}$. The y-axis is logarithmic.

## 5 The 2D implementation

All of the previously mentioned features are also implemented in the 2D version of the model. We consider a configuration with a homogeneous host rock containing a weak inclusion to perturb the stress field and initiate localization (Fig. 2b). In Fig. 8, we show the temporal evolution of such a 2D simulation, using the same parameters as the 1D reference model and a regularization viscosity of $\eta_{\mathrm{reg}} = 10^{12}$ Pa·s.

The 2D model undergoes the same stages as in 1D. An initial, homogeneous elastic loading stage is followed by the onset of LTP at the tips of the inclusion. Subsequently, a shear zone forms and starts to develop horizontally across the domain (Fig. 8a, b), before deformation becomes more localized near the anomaly tips (Fig. 8c). Thermal runaway initiates here and then propagates horizontally across the domain (Fig. 8d-f), creating a rupture front marked by a sharp stress gradient (Fig. 8, left column) and a peak in horizontal velocity (Fig. 8, central column). The simulation is stopped once the stress is fully released. Here, we focus on the numerical behavior of the 2D model; for a detailed discussion of the physical implications, refer to Spang et al. (2025a).

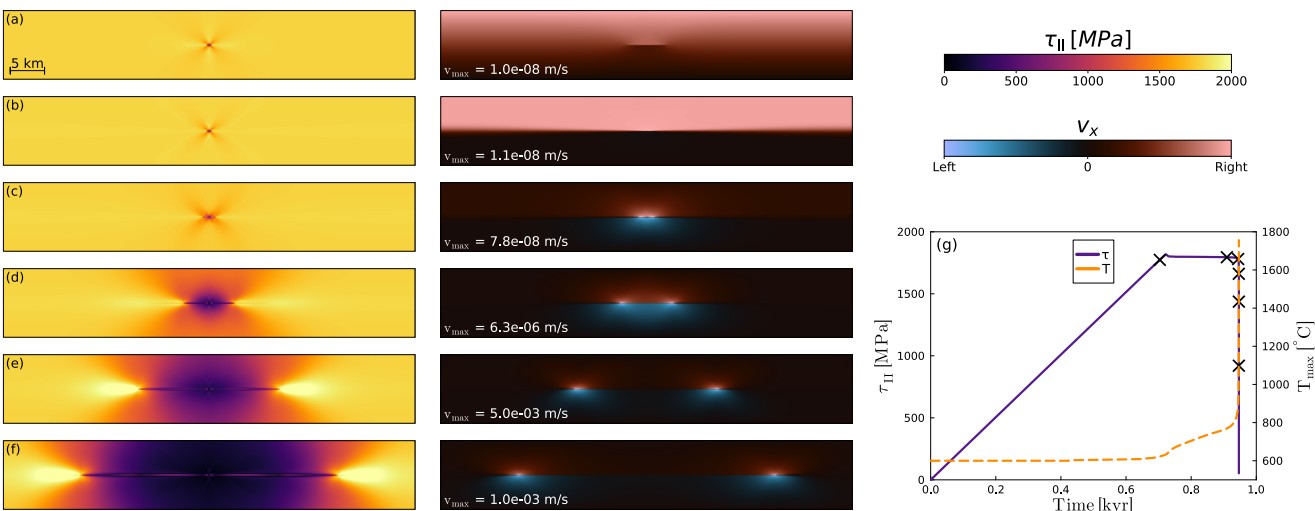

**Figure 8.** Thermal runaway in 2D. (a-f) Temporal evolution of stress (left) and horizontal velocity (center) fields. (g) Temporal evolution of average stress and maximum temperature. Black crosses along stress curve indicate the six snapshots shown in (a-f). The model uses the parameters given in Table 1, with the exception of $\eta_{\mathrm{reg}} = 10^{12} \, \mathrm{Pa \cdot s}$.

## 5.1 Role of solution strategies in 2D

Adaptive time stepping remains critical in 2D. During elastic loading, time steps are typically on the order of decades; they shrink to months at the onset of thermal runaway, to hours during rupture propagation, and to seconds at peak velocities. Setting a lower time step bound can dampen thermal runaway, or, if set too high, cause solver failure. For most of the simulations, the predictive time stepping strategy (Sect. 4.1.1) suffices. However, when rupture fronts meet across the periodic boundaries, restarting time steps (Sect. 4.1.3) is required to maintain stability.

Regularization plays a similar role in 2D as in 1D. It enforces a lower bound on viscosity and upper bounds on strain rate and velocity. Due to the more limited spatial resolution in 2D, the shear zone thickness is often constrained by grid size unless a high regularization viscosity ($\sim 10^{16}$ Pa·s) is used. If a higher spatial resolution can be achieved through improved refinement or significant increase in grid cells, regularization viscosity will again become the controlling factor. This equally applies to

adaptive rescaling (Sect. 4.2), which becomes essential when smaller time steps and higher velocities exacerbate round-off errors. Given its superior performance at fine resolutions, the viscosity regularization is the preferred method in 2D.

Finally, monitoring the convergence of the relaxed viscosity (Sect. 4.4) has minimal impact in 2D. Even before reaching the LTP threshold, the number of iterations per time step increases to $\sim 5000$ to solve the conservation equations, ensuring that the relaxation-based updates are well-converged.

## 5.2  Comparison to 1D

To compare the 1D and 2D models, we ran a 1D model using the same limited refinement as in 2D and compared the results (Fig. 9). Both models exhibit very similar trends in stress, maximum temperature, maximum velocity, and minimum viscosity. As long as the thermal runaway fronts remain more than 10 km away from the domain boundaries (Fig. 8a-f), the 2D and 1D models show nearly identical $v_{\max}$ and $T_{\max}$ (Fig. 9b,c). Once the rupture fronts meet due to periodic boundaries, $v_{\max}$ increases by a factor of $\sim 3$ and $T_{\max}$ by $\sim 250\,^{\circ}\text{C}$. This surplus is likely caused by the increased stress in front of the rupture tips (yellow lobes in Fig. 8). When the tips connect, they can release more stress which is converted into heat, resulting in faster slip.

The most notable difference is the duration of the LTP-dominated phase, which lasts over 10 kyr in the 1D model but only $\sim 200$ years in 2D. This discrepancy stems from differences in how the anomaly is defined. In 1D, only the flow laws of diffusion and dislocation creep are weakened. In 2D, the LTP back stress $\sigma_{\text{b}}$ is also reduced. This difference was necessary as reducing $\sigma_{\text{b}}$ in 1D prevents stresses in the entire model from reaching values above 1 GPa, while omitting this weakening in 2D hampers localization significantly.

## 6  Simplifications and design choices

### 6.1  Governing equations

As stated above, we neglect inertial terms from Eq. (1) for simplicity. To determine to which extent this assumption is justified, we roughly estimate the inertial term $\rho \frac{\partial v_i}{\partial t}$. In 2D, we assume the extreme case that a grid node accelerates to the maximum velocity ($5\,\text{mm}\,\text{s}^{-1}$) in a single time step ($\sim 15\,\text{s}$). The resulting value for $\rho \frac{\partial v_i}{\partial t}$ is on the order of $1\,\text{kg}\,\text{s}^{-2}\,\text{m}^{-2}$, whereas the term $\frac{\partial \tau_{ij}}{\partial x_j}$ on the right hand side of the governing two-dimensional momentum equation is about seven orders of magnitude larger. In this case, neglecting the inertial term remains justified. However, in 1D models with the lowest tested values of $\eta_{\text{reg}}$ or $\lambda_{\text{reg}}$, the inertial term could reach much larger values due to the larger velocities. In this case, inertia could reduce acceleration.

Gravity is neglected from Eq. (1) because the orientation of the shear zone is arbitrary in reality. Thermal expansion is neglected from Eq. (4) and adiabatic heating from Eq. (3) as they did not play a significant role in a previous 2D study on thermal runaway (Spang et al., 2025a).

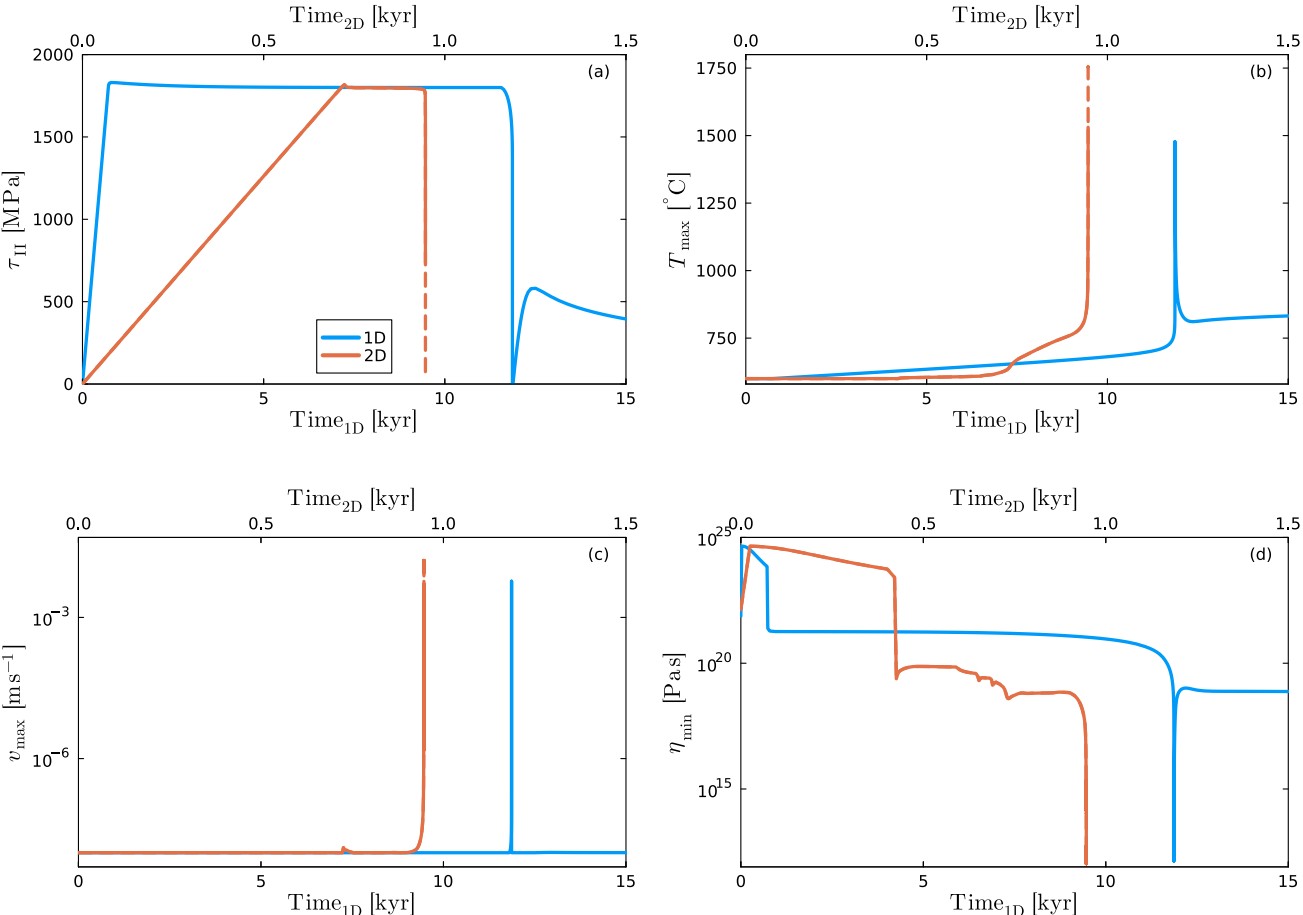

**Figure 9.** Comparison between 1D (blue) and 2D (orange) simulations. Note the different x-axes. Dashed lines indicate the portion of the 2D model influenced by periodic boundary conditions. (a) Mean deviatoric stress. (b) Maximum temperature. (c) Maximum velocity. (d) Minimum viscosity. Both models use the parameters given in Table 1, with the exception of $\eta_{\text{reg}} = 10^{12}$ Pa·s. 1D model uses the same vertical grid spacing as the 2D model (Sect. 3.1).

## 6.2 Grain size evolution

Adding grain size evolution could have a significant impact on the rheology and energy balance of our models, depending on how much energy from viscous dissipation is partitioned into it. The partition factor spans several orders of magnitude in the literature with a maximum of 10% (e.g., Mulyukova and Bercovici, 2017; Ruh et al., 2024). Furthermore, it might be strain-dependent, as experimental studies suggest that only about 10% of olivine grains recrystallize at a strain of 1 (Cross and Skemer, 2019). Most of our models do not even reach a strain of 0.1.

## 6.3 Phase transformation

Endothermic phase transformations are another potential sink for thermal energy during runaway. Intermediate-depth earthquakes are commonly associated with the antigorite-olivine transformation (Hacker et al., 2003), and Brantut et al. (2017) estimate the enthalpy change of this reaction to be on the order of $2.5 \cdot 10^8$ J m$^{-3}$. This is about one quarter of the energy density required for full melting. Consequently, this process would not have a significant effect on the energy balance during thermal runaway. Deep-focus earthquakes are associated with the olivine-ringwoodite transformation (Kirby et al., 1996), but this reaction is exothermic (Gleason and Green II, 2009) and cannot act as an energy sink during runaway.

## 6.4 Model setup

We only show cases with a single perturbation. As the 2D setup uses periodic boundary conditions, it approximates a setup with multiple perturbations on the same vertical coordinate. The results show that temperature and slip velocity peak when two rupture fronts unite (Fig. 9b,c). A more realistic case could involve perturbations of different size, strength, and location. Comparing the length of the LTP-dominated warm-up period with the runaway phase suggests that once runaway initiates in one location, the rupture would quickly release stress from surrounding perturbations, resulting in a single dominant rupture.

## 6.5 1D results

Figure 5b illustrates that 1D models can reach temperatures that exceed any observed or constrained values for the Earth when using viscosity regularization. Similarly, models using gradient regularization reach slip velocities that are significantly faster than any observed solid deformation, including earthquake slip and seismic waves (Fig. 5e). These unrealistic values are inherent to 1D models as they imply an infinite shear zone (e.g., Kameyama et al., 1999; Braeck et al., 2009). While such models struggle to accurately describe peak runaway conditions, they are still useful in investigating how localization develops in the first place (e.g., Ogawa, 1987; Thielmann et al., 2015; Spang et al., 2024).

## 7 Conclusions

Resolving strain localization owing to thermal runaway represents a numerical challenge due to its spontaneous onset, rapid self-acceleration, extreme localization, and strong gradients in temperature and viscosity. We address these by implementing adaptive time stepping based on changes in stress and temperature and allowing time steps to be restarted if necessary. We achieve a time step reduction by more than ten orders of magnitude without destabilizing the solver. To maintain numerical precision during such extreme changes, we rescale time-dependent properties using an adaptive internal time scale.

To handle the self-localizing nature of thermal runaway, prevent solver failure from excessive viscosity reduction, and keep results reproducible, we introduce regularization. Viscosity and gradient regularization both limit maximum velocity and temperature and impose a minimum shear zone width, without altering the overall stress evolution. Viscosity regularization

more strongly constrains velocity, whereas gradient regularization better controls temperature increase and shear zone width. Accounting for the latent heat of melting or phase transformations is not sufficient to regularize thermal runaway.

We also show that the commonly used viscosity relaxation method in pseudo-transient schemes can result in incorrect stress evolutions near the LTP threshold. Only accepting solutions with a sufficiently converged viscosity ensures accurate stress evolution.

Extending the model to two spatial dimensions preserves the key physical behavior observed in 1D. Although 2D simulations are more limited in spatial resolution due to grid aspect ratio constraints, adaptive time stepping, regularization, and rescaling remain essential. Since 2D models naturally require more iterations per time step, monitoring viscosity convergence is less critical.

*Code and data availability.* The current version of DEDLoc (Deep Earthquake Ductile Localization) is available from the project website https://github.com/ArneSpang/DEDLoc under MIT licence. The exact version of the model used to produce the results used in this paper is archived on Zenodo under https://doi.org/10.5281/zenodo.15481111, as are input data and scripts to run the models for all the simulations presented in this paper (Spang et al., 2025b).

*Video supplement.* A video of the 2D model is available on Zenodo under https://doi.org/10.5281/zenodo.15481111 (Spang et al., 2025b).

## Appendix A:  Strain rate partitioning

The solver consists of 6 repeating steps:

1. Compute full strain rate from velocity field

2. Partition strain rate among elasticity, diffusion creep, dislocation creep, low-temperature plasticity, and the regularization

3. Compute the viscosity of each individual mechanism

4. Compute effective viscosity

5. Compute stress

6. Update velocity, pressure, and temperature

Step 2 is especially challenging, so we present our strategy here. Figure A1 illustrates our rheological model including viscosity regularization. The main challenges are the partition of stress between the regularization branch (orange in Fig. A1) and the viscous branch (blue in Fig. A1), as well as the partition of the viscous strain rate between the different mechanisms. Stress is equal in sequential components and partitioned in parallel components, strain rate vice-versa (Maxwell, 1867; Jóźwiak et al., 2015). For clarity, we have neglected the subscript $_{II}$ in the following equations.

First, we partition the strain rate between the elastic and viscous / regularization components. The elastic strain rate can be expressed as follows:

$$\dot{\varepsilon}_{\text{el}} = \frac{\tau - \tau^{\text{old}}}{2G\Delta t}, \tag{A1}$$

where $\tau$ refers to the current stress and old refers to the stress at the end of the previous physical time step. This allows us to compute the viscous strain rate:

$$\dot{\varepsilon}_{\text{vi}} = \dot{\varepsilon} - \dot{\varepsilon}_{\text{el}}. \tag{A2}$$

$\dot{\varepsilon}_{\text{vi}}$ is identical in the viscous branch and the regularization branch, and since $\eta_{\text{reg}}$ is known, we can express the stress carried by the regularization as follows:

$$\tau_{\text{reg}} = 2\dot{\varepsilon}_{\text{vi}}\eta_{\text{reg}}. \tag{A3}$$

As stress is partitioned between the viscous and regularization branch, we can compute the viscous stress by:

$$\tau_{\text{vi}} = \tau - \tau_{\text{reg}}. \tag{A4}$$

Viscous stress is identical in all viscous components, but viscous strain rate is partitioned between them. As diffusion creep viscosity is independent of the partitioning, the diffusion creep component can be computed by:

$$\dot{\varepsilon}_{\text{dif}} = \frac{\tau_{\text{vi}}}{2\eta_{\text{dif}}}. \tag{A5}$$

$\dot{\varepsilon}_{\text{dif}}$ can be subtracted from the viscous strain rate to find the nonlinear part which partitions into dislocation creep and low-temperature plasticity.

$$\dot{\varepsilon}_{\text{nl}} = \dot{\varepsilon}_{\text{vi}} - \dot{\varepsilon}_{\text{dif}} = \dot{\varepsilon}_{\text{dis}} + \dot{\varepsilon}_{\text{LTP}}. \tag{A6}$$

If neither dislocation creep nor LTP are currently active (i.e. taking a significant strain rate partition), $\dot{\varepsilon}_{\text{nl}}$ can become negative. In this case, we overwrite it with a very small positive value as a negative value or zero would cause issues in the viscosity calculation.

As $\eta_{\text{dis}}$ and $\eta_{\text{LTP}}$ both depend on the strain rate partitioning, we can not solve for either strain rate component analogously to Eq. (A5). But, since $\dot{\varepsilon}_{\text{dis}}$ and $\dot{\varepsilon}_{\text{LTP}}$ are inversely proportional to $\eta_{\text{dis}}$ and $\eta_{\text{LTP}}$ respectively, we can guess their ratio from the viscosities of the previous iteration.

$$\frac{\dot{\varepsilon}_{\text{dis}}}{\dot{\varepsilon}_{\text{LTP}}} \approx \frac{\eta_{\text{LTP}}^{\text{prev}}}{\eta_{\text{dis}}^{\text{prev}}} = r_\eta \tag{A7}$$

This yields:

$$\dot{\varepsilon}_{\text{dis,g}} = \dot{\varepsilon}_{\text{nl}} \frac{r_\eta}{1 + r_\eta}, \tag{A8}$$

$$\dot{\varepsilon}_{\text{LTP,g}} = \dot{\varepsilon}_{\text{nl}} \frac{1}{1 + r_\eta}, \tag{A9}$$

where $\dot{\varepsilon}_{\text{dis,g}}$ and $\dot{\varepsilon}_{\text{LTP,g}}$ are guesses for the strain rate of dislocation creep and low-temperature plasticity respectively. $\eta_{\text{dis}}$ and $\eta_{\text{LTP}}$ are computed with these guesses according to Eq. (12) and (13), and after stress has been updated, the true partitioning for both mechanisms can be computed analogously to Eq. (A5):

$$\dot{\varepsilon}_{\text{dis}} = \frac{\tau_{\text{vi}}}{2\eta_{\text{dis}}}, \tag{A10}$$

$$\dot{\varepsilon}_{\text{LTP}} = \frac{\tau_{\text{vi}}}{2\eta_{\text{LTP}}}. \tag{A11}$$

During the pseudo-time iterations, $\dot{\varepsilon}_{\text{dis,g}}$ and $\dot{\varepsilon}_{\text{LTP,g}}$ converge towards $\dot{\varepsilon}_{\text{dis}}$ and $\dot{\varepsilon}_{\text{LTP}}$ respectively. We track this convergence and use it as an additional requirement for a solution to be accepted. If gradient regularization is used, the orange component in Fig. A1 is missing, and $\tau_{\text{vi}} = \tau$.

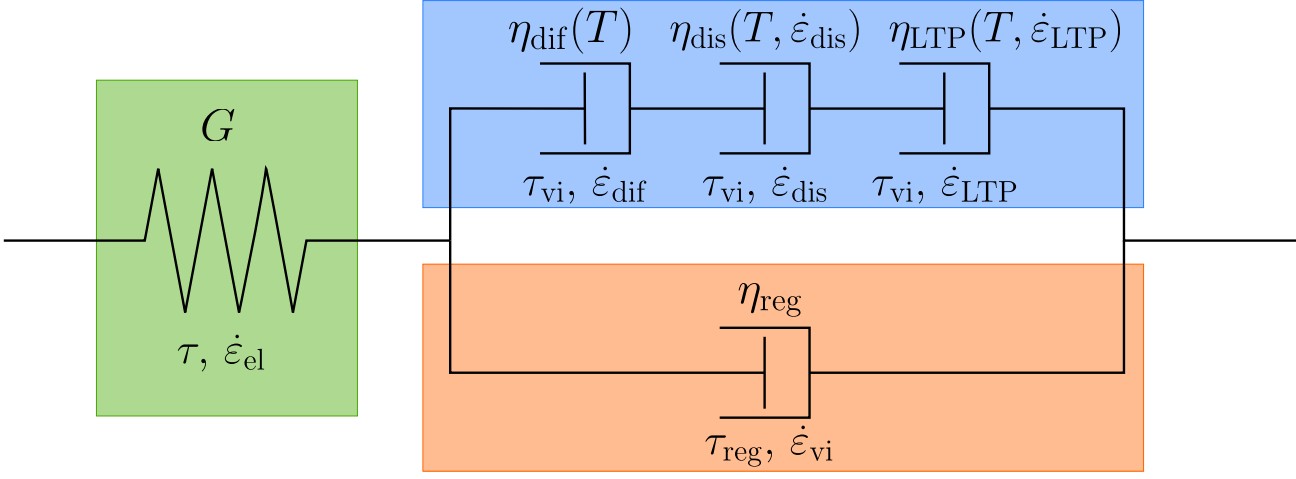

**Figure A1.** Illustration of our rheological model including the viscous regularization. Green shaded region shows elastic component, blue shows viscous component, and orange shows regularization component. Individual deformation mechanisms are labeled with their respective stresses and strain rates.

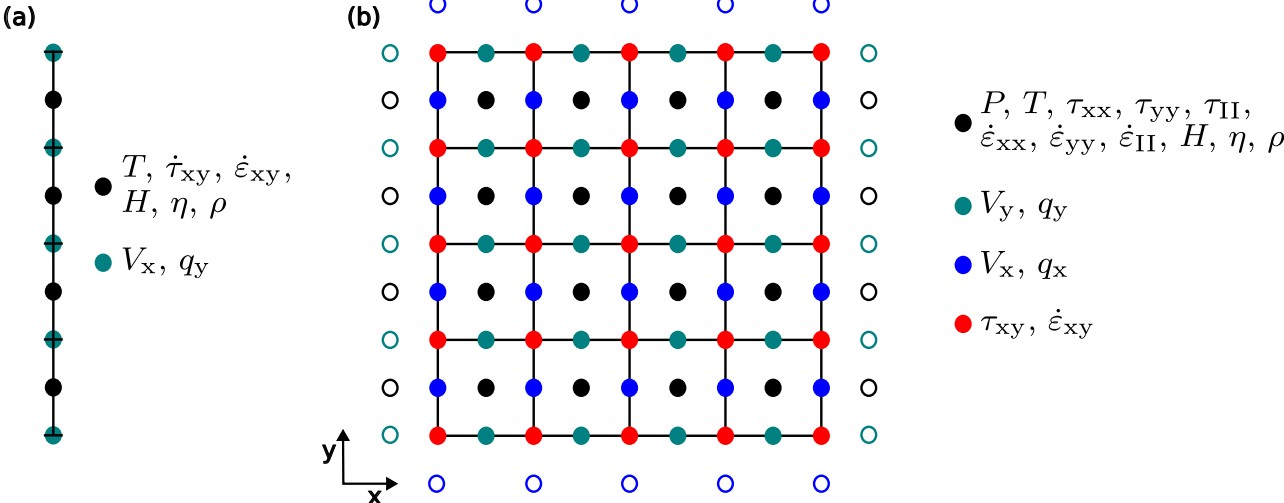

**Figure A2.** Illustration of staggered numerical grid, indicating where different parameters are computed. (a) 1D. (b) 2D. Hollow circles are ghost nodes outside the physical domain which are necessary to employ boundary conditions. Modified from Spang et al. (2025a).

*Author contributions.* A.S. was the main developer of the code and methodology of the study, ran all simulations, processed the results, produced the Figures, wrote the original manuscript and edited it after co-author revision. M.T. contributed to the methodology, provided the funding and reviewed the manuscript. C.P. provided the theory for the gradient regularization. A.d.M and L.R. helped with the implementation of the pseudo-transient method, reviewed and edited the manuscript.

*Competing interests.* L.R. is a member of the editorial board of Geoscientific Model Development.

*Acknowledgements.* A.S. and M.T. were funded by the DFG grant TH 2076/8-1 awarded to M.T.. This research has been supported by the Swiss University Conference and the Swiss Council of Federal Institutes of Technology through the Platform for Advanced Scientific Computing (PASC), obtained via the PASC project GPU4GEO. The authors thank Laetitia Le Pourhiet, Cedric Thieulot, and one anonymous reviewer for their constructive comments.

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
