# Peer review of "Overcoming the numerical challenges owing to rapid ductile localization with DEDLoc (version 1.0.0)"

_EGUsphere, 2025_

## Author Comment (AC2)

**General Assessment**

The paper by Spange et al. compares different methods to regularize ductile localization associated with thermal runaway in the ductile regime. This is a challenging and important issue for the community, since the length scales of shear zones produced by this process in the Earth's mantle may be on the order of nanometers to millimeters and evolve over seconds, while models typically operate at scales of hundreds of kilometers and millions of years. Regularization is therefore essential.

The paper is well written and the results are presented clearly. The main weakness, in my opinion, is the lack of separation between the physical aspects and the numerical methodology in Sections 2 and 3. I believe this could be addressed by a modest reorganization of the text.

**Broader Comments**

- The paper is primarily methodological, but a stronger discussion of the underlying physics would be very valuable. In particular, it would help to reflect on how the approximations in the heat equation, mass conservation, boundary conditions, and the assumption of a single shear band simplify or complicate the problem compared to more Earth-like conditions.

    **Reply:** We have added a new section (6) to discuss the simplifications and design choices regarding inertia, gravity, thermal expansion, adiabatic heating, and grain size evolution. The section also discusses the case of multiple perturbations that could act as nucleation sites for thermal runaway.

    Regarding thermal boundary conditions, they have no impact during the runaway as the heating rate in the shear zone is orders of magnitude faster than thermal diffusion. In fact, once runaway starts, heating close to the boundaries essentially ceases as all dissipative work occurs in the center. Using constant temperature boundaries would increase runaway intensity as the host rock would be cooled during the LTP-phase, increasing the viscosity contrast between host rock and shear zone. We are attaching the results of a 1D model below.

    Regarding the periodic boundary conditions, we can create a similar result with pure shear boundary conditions. In this case, the shear zone forms at an angle of 45°.

[Figure]

Maximum and minimum (inset) temperature for no-flux boundary condition (blue, used in our study) and constant temperature boundary condition (orange) in a 1D model. Temperature is saved at cell centers, so even for constant T boundaries, there is a little bit of increase in T_min.

- It would also be interesting to discuss whether the length scales or gradients introduced by the regularization might have a physical meaning. Ideally, the problem should be governed by material parameters rather than numerical parameters. While the proposed regularizations alleviate mesh dependence, they introduce sensitivities to new parameters that are equally numerical. Introducing physically motivated mechanisms that could act as natural regularization would strengthen the study.

  **Reply:** We agree that physically motivated regularizations or, ideally, actual physical mechanisms would be more satisfying. However, such processes can introduce new uncertainties if they are not constrained by sufficient experiments. Furthermore, adding physical processes can add more non-linearities to the system and potentially make the numerical solution even less stable. We introduced a new subsection (4.3.5) to discuss the relation of both regularization approaches to physical processes, the value of adding more physical processes, and the potential problems related to these additions. Furthermore, we added a third regularization method with the latent heat of melting based on the suggestions of reviewer 3.

**Suggestions for Reorganization**

1. Introduce the governing equations (Sec. 2.2.1), rheology (Sec. 3.2), and density (Sec. 3.3) **before** describing the model setup (Sec. 2.1) and the 1D simplifications (Sec. 2.2.2).

   **Reply:** We adopted this order for section 2.

2. In Section 3, begin with the spatial discretization, then proceed through the pseudo-transient scheme, nonlinear viscosity iterations, and finally regularization.

   **Reply:** We adopted this order for section 3.

3. Please add the thermal boundary conditions to the description of the model setup.

   **Reply:** We have added the thermal boundary conditions to the beginning of the model setup section.

**Notation and Tables**

- Since you use $\lambda$ as a regularization length scale, I suggest using $\kappa$ for heat diffusivity (instead of $\lambda$). Consequently, use $k$ for heat conductivity in Eq. 17 and in the text (L.122), and update the last line of Table 1. Currently, you list conductivity but label it as diffusivity.

  **Reply:** We agree that using $\lambda$ and $\lambda_{reg}$ with different units is inconsistent. We have therefore replaced $\lambda$ (thermal conductivity) by k.

- In Table 1, group the elastic bulk modulus $K$ with the shear modulus $G$. It does not need the subscript "b". You could compute $K$ using Poisson's ratio of 0.25 and then remove Poisson's ratio from the table, along with Eq. 28. For an isotropic elastic material, only $K$ and $G$ are needed; Poisson's ratio can simply be mentioned as an explanatory note.

  **Reply:** We added bulk modulus to Table 1, next to shear modulus, and removed Poisson ratio. We now note that bulk modulus is computed from G and Poisson ratio at the bottom of the table and removed Eq. 28. We decided to keep the subscript "b" to avoid potential confusion with the new symbol for thermal conductivity (k).

**Technical and Line-by-Line Comments**

- **L.23:** While plate-scale models may overestimate shear zone width due to coarse resolution, could you add a reference or explanation for grain-scale models potentially underestimating them? If Braeck et al. are correct, shear zones could be as thin as nanometers—smaller than grain-scale models can capture.

  **Reply:** This section was poorly worded. We split the sentence in two and extended the second part to explain that a grain scale model might be too small cover the relevant geological context to accurately compute the size of the shear zone.

- **Eq. 27:** Please provide a brief explanation of how it follows from the mass conservation equation. I suggest moving this equation (with the explanation) next to Eqs. 2 and 4, without devoting a full subsection to density.

  **Reply:** Agreed. We moved this equation to the governing equations section and added that it follows from the combination of equations 2 and 4 and the integration of the changes of pressure and density. We then removed the density subsection.

- **Eq. 3:** Explicitly note that $\tau_{ij} e_{ij}^{vi}$ is the shear heating term, and that all dissipated work is assumed to convert into heat (i.e., no grain size reduction).

  **Reply:** We added this clarification below equation 3.

- Consider placing remarks on neglected terms close to the relevant equations:

  - inertial and gravity terms after Eq. 1,

  - adiabatic and radiogenic heating after Eq. 3,

  - thermal expansion after Eq. 4.

  **Reply:** We would like to keep the equations in one block. To improve clarity, we now mention which equations the neglected terms belong to.

- Use $e$ instead of $\dot{\varepsilon}$ \dot{\varepsilon} for the deviatoric strain rate, consistent with your notation for deviatoric stress ($\tau$ vs. $\sigma$). Also, define $\tau$ explicitly, just as you do for the strain rate tensor.

**Reply:** We would like to stick with the use of \dot{\varepsilon}. We do not mention volumetric strain rate or full strain rate anywhere in the text and clearly state how we define \dot{\varepsilon} in equations 5 and 6 (now 6 and 7) and the lines in between. We also think the use of e is not ideal as it can be mistaken for the Euler number. One other common notation is \dot{\varepsilon}' but this is not ideal when specifications like viscous or dislocation are added in the exponent as we do in equation 18 (now 8). The accepted papers of Popov et al. and Li et al. in this issue also use \dot{\varepsilon} for deviatoric strain rate.

- **L.84:** effective shear viscosity (clarify wording).

  **Reply:** We added this clarification.

- **L.89:** Gravity does not need to be listed here, as it does not affect Eqs. 7–8 if initial stress conditions already include gravity and they should.

  **Reply:** Gravity is not taken into account for the initial stress conditions. The initial stress is zero in the entire domain. We list it because if we were taking gravity into account, divergence of velocity would not necessarily be equal to zero as material would compact due to gravitational forces.

- **L.97:** Since you previously called them "deviatoric," maintain that wording consistently.

  **Reply:** In order to not clutter the text in the following sections, and since we never discuss volumetric stress or strain rate, we instead added a statement at the end of the governing equations section that all following references to stress and strain rate refer to the deviatoric components.

- **Eq. 17:** Replace λ with κ

  **Reply:** All instances of λ were replaced by k.

- **L.123:** Use k=κ/(ρCp).

  **Reply:** We now use κ =k /(ρCp) in line with our change from λ to k.

- **L.145–150:** Please separate physics from numerics; for instance, move stabilization viscosity details to a new subsection of Section 3.

  **Reply:** We split this part of the rheology section from the physical part and it is now the section "3.3 Viscosity update"

- **L.203:** Clarify why τ is used instead of change in strain rate? If this comes from the cited reference, a short explanation in the text would help.

  **Reply:** This decision does not come from the cited references. We chose temperature and stress change as thermal runaway is driven by the conversion of elastic energy (i.e. stress) to thermal energy (i.e. temperature). We included this reasoning to the manuscript.

- **L.260–265:** A short note on how elasticity may improve conditioning would be useful.

  **Reply:** The comment appears to imply that we do not use elasticity, but we do. Elasticity acts as a buffer when stress conditions change rapidly. In areas of low viscosity, it limits the magnitude of deformation. We added a sentence about this to the section in question.

- **L.355/L.375:** You mention using the same parameters in 1D and 2D, but then describe differences in how the anomaly is defined. Please clarify.

  **Reply:** The parameters are the same. However, there is a small difference in how the weakening parameter ω was applied. This is also described in the Model setup section. The difference in applying ω was necessary due to the geometry of the 1D model. In 1D, weakening the LTP flow law means that the entire 1D model cannot reach 1.8 GPa, due to the shear stress being constant throughout the model. In 2D, only the small inclusion cannot reach this stress while the rest of the domain can. We have restructured section 5 which should help clarify this issue.

- **L.370:** A comment on the effect of periodic boundary conditions would be helpful. In principle, once the shear band spans the entire domain, 1D and 2D solutions should converge and not diverge as 1D is an infinite shear band.... I don't really understand.

**Reply:** Once the rupture spans the entire 2D domain across periodic boundaries, it indeed represents an infinite shear band like the 1D model. However, there are still differences between the two settings. In 1D, the entire shear zone is represented by a weakened rheological flow law whereas in 2D, only a very small portion of the shear zone (the initial anomaly) has a perturbed flow law.

What is probably more important is the heterogeneous stress field in 2D (Fig. 8, left column). The rupture tips increase the stress ahead of themselves. In the 1D model, runaway releases 1.8 GPa of stress. In 2D, the average stress across the model domain is the same, but in the stress lobes ahead of rupture tips, the stress exceeds 2 GPa. When the two tips unite, they have more stress (i.e. energy) to release than in the 1D example and consequently reach larger temperature and slip velocity. We added this explanation to the section in question.

Once the stress has been fully released in the 2D model, we would expect it to be very similar to the 1D case again.

- **L.382:** When the timestep drops to seconds, and given the large stress drops you show, is it still valid to neglect inertial terms? Could you add a small comment on that?

  **Reply:** It is correct that inertia may start to play a role at peak runaway conditions and reduce accelerations.

  For 2D, we can make a simple estimation of the inertia term. Figure 8e shows the peak velocities during the runaway. For an upper limit estimate, we assume that a grid node accelerates from stationary to peak velocities (5 mm/s) in one time step (15 s). With a density of about 3300 kg/m^3, this yields about 1 kg/(m^2 s^2). As a comparison, the stress gradient term reaches up to 10^7 kg/(m^2 s^2). So, inertia should still be negligible.

  In our 1D models, we cannot make this comparison as the stress gradient is always zero following the momentum conservation. For cases with very low eta_reg or lambda_reg, the inertia term would also be larger than 10^6 kg/(m^2 s^2). Here, it should have a significant impact. We added these considerations in the new section 6.1.

**Recommendation**

I recommend **minor revisions**, focused mainly on clarifying the separation between physical and numerical aspects, improving the discussion of the physical meaning of

regularization, and addressing the minor notational and organizational issues listed above.

Laetitia Le Pourhiet

**Reply:** We thank the reviewer for their encouraging words and constructive criticism. We have adopted the proposed separation between physics and numerics. Furthermore, we have added a discussion of the physical effects of the regularization techniques alongside other small corrections.

---

## Author Comment (AC3)

I have read with interest the submitted manuscript by Spang et al. which builds on Spang et al 2024. I found it well written, well organized, and the goal of the paper is also properly stated and addressed convincingly. I here wish to commend the authors for their didactic approach (e.g. the detailed section 3, and the Appendix) which helps the reader to make sense of the rather technical presented research without having to necessarily first read the previous article on the topic by the same authors.

**Reply:** We thank the reviewer for the encouraging words.

Furthermore, because of the review process of this journal I have the luxury to have access to the thorough review of Prof. Le Pourhiet and I agree with all her points. As such I will keep my review very short and also recommend minor revisions.

**Reply:** We posted our reply to Prof. Le Pourhiet, so we will not copy them here.

My only point of criticism is the fact that the 2d case presented in the paper is very simple (which of course allows for testing the various approaches) but I feel a more 'realistic' case could be added (for example a simple crustal model which does not rely on periodic boundary conditions?).

**Reply:** We agree that our 2D example is very simple. Our aim for this paper is to explain how rapid ductile localization can be dealt with numerically. As such, we don't think an application case fits the scope of the paper. Furthermore, the focus of our study is thermal runaway as a potential driver of deep earthquakes, therefore a crustal model would not be appropriate in our opinion. A more appropriate setup would include a subducting slab in the mantle transition zone where the model domain of our 2D example would be aligned with the slab. This would require a larger domain site and thus significantly limit the resolution of the shear zone. It would also require oblique boundary conditions to induce a deformation field that is comparable to the 1D simple shear case. We therefore think, a more complex model setup is out of the scope of this study. Regarding the periodic boundary conditions, similar results can be obtained with pure shear boundary conditions. In this case, the shear zone forms at an angle of 45°.

Finally, a minor problem: at line 64 the authors state that "Adif and Adis are multiplied by 2, and sigma_b is divided by 2". Adif and Adis are defined a few lines above, but not sigma_b at that stage of the paper.

**Reply:** We followed the suggestions of Reviewer 1 and reorganized sections 2 and 3. Line 64 now comes after the rheology section, so sigma_b will be defined before it appears in the model setup description.

---

## Author Comment (AC4)

The study by A. Spang and co-authors investigate the numerical issues for the problem of ductile strain localisation and propose several techniques how to mitigate them. The investigation is clear and systematic, however, the main weakness is that all the proposed methods (maybe less the gradient regularisation) to overcome the problems are numerical in nature, and are less grounded in the physics of the problem. For example, in the majority of the tests, the temperature during thermal runaway reaches 5000 K or more (which is ~surface temperature of the Sun and a lot higher than any current estimates for the core-mantle boundary temperature) - and I'm surprised nowhere in the manuscript this is discussed that it's not physical. When temperatures reach the solidus (between 1000-2500 K for most rocks in the upper mantle), rocks starts melting. During melting, energy in the form of latent heat is consumed, and temperatures stay constant. Have the authors considered a stabilisation mechanism based on temperature (i.e., limit Tmax) and the thermodynamics of melting during strain localisation? This could be a starting point on how to regularise/limit the runaway from a physics angle.

**Reply:** We agree that the 1D models reach unrealistic values (temperature with viscosity regularization, velocity with gradient regularization). The reason for this is that 1D models assume an infinite shear band. These models are inherently unrealistic and produce unphysical results in the end-stage of a self-feeding process like thermal runaway. The same result is observed in similar studies like Kameyama et al., 1999, Figure 3 (model is stopped at reaching 1200 K after only 1% of stress drop) or Braeck et al., 2007, Figure 4 (non-dimensionalized temperature and clipped colormap). We did not comment on the values as they are not the focus of the study (numerical stability is) and common for these kinds of models. But we agree that a statement is helpful and added a new paragraph with this explanation (new section 6.5).

We have added a form of melting as a third regularization technique to our code. We use the anhydrous parameterization of Katz et al., 2003 to compute melt fraction and take latent heat into account in the energy conservation equation (new section 3.4.3). Melting should introduce additional weakening in the rheology and increase the intensity of thermal runaway, but we neglect this for simplicity. It is essentially a best-case scenario for melting as a regularization.

The results are shown in the new Figure 6 and section 4.3.3. The shear zone (1D) or rupture tip (2D) melts completely within milliseconds and then temperature continues to grow. Melting cannot regularize thermal runaway. We also added a paragraph to section 4.3.4 to discuss what mechanisms would be necessary on top of melting to make it more effective energy sink.

Limiting the maximum temperature without invoking a physical mechanism would mean artificially creating an energy sink and violating energy conservation. We think

that this is not more physical than our methods and that it provides less control than viscosity regularization.

Therefore, I suggest that the authors discuss their regularisation methods more in terms of physics. Of course numerical stability and solver convergence are important, but the choices for regularisation need to be justified more. This would highly improve their study, and make the proposals more useful to the scientific community. The manuscript is mostly well-written and well-structured, although it requires some revision of the text to avoid vague statements. I detail these points below. Considering that the revision work I propose is considerable, I recommend a major revision of the manuscript, but could be a valuable contribution to GMD.

**Reply:** We agree that the different regularization methods should be put in more context with respect to potential underlying physical mechanisms. We added a new section (4.3.5) to discuss physical interpretations of the regularization approaches we present.

However, our study focuses on the numerical aspects of employing different regularization approaches and we see this study in line with other studies (Jacquey and Cacace 2020, Duretz et al., 2023, Goudarzi et al., 2023, all cited in the manuscript) that also focus on the numerical aspects. We also want to stress here that we do not claim that the proposed numerical approaches represent the only solution to this problem.

1. What is the physical explanation for the proposed methods? I think they need to be better justified.

* This starts with the title. The authors changed the title after submission to include the code name. I suggest they revert to something close to the original title "Overcoming numerical challenges during rapid ductile localisation", because the information in this manuscript should transcend the numerical implementation hopefully (despite faster future languages and algorithms the lessons here will still hold for the problem of ductile strain localisation). The authors use DEDLoc as a tool to demonstrate the proposed techniques.

**Reply:** We agree with the reviewer here. It is exactly the reason why we did not include DEDLoc in our original manuscript title. However, it is journal policy that the code name needs to be included in the title, therefore we had to add the code name. We would be happy to revert to the original title if given the possibility.

* The viscosity regularisation (Section 3.5.1) is essentially a minimum viscosity cut-off technique. However, it can be thought as another flow law defining an isotropic

viscosity, which is active together with ductile flow laws. My point is that the authors should not introduce "extra-bits" in the physical model that are not justifiable.

**Reply:** The viscosity regularisation is indeed a minimum viscosity cut-off. The analogy to another flow law active alongside diffusion creep, dislocation creep, and LTP is however not quite accurate. As the regularisation viscosity is in parallel with the other flow laws (Figure A1), it is only active if one of them becomes as weak as the regularization. At this point, the flow laws might not be accurate anymore anyways.

Regarding adding "extra-bits" to the physical model, there might be a misunderstanding. We are not claiming anywhere that these are physical. We are not trying to mimic a physical process with viscosity or gradient regularization. We are trying to achieve a reproducible numerical result that is physically accurate until the regularization starts to play a role. And beyond that point, it is clear which numerical parameters govern the model behavior and in what way. As such, these "extra-bits" are justifiable and necessary in our opinion.

* While the gradient regularisation (Section 3.5.2) seems more physical, the significance of lamda_reg is not explained. For example, what to expect with shear heating and localisation when lambda_reg is small/large?

**Reply:** We added a sentence to this section to explain that a larger lamda_reg results in more smoothing and damped runaway.

* All the methods controlling time-step size and the implementation (within/outside solver iterations) need to be explained how they impact the physics (i.e. stress states, especially since dt impacts elastic stresses). For example, even the Courant–Friedrichs–Lewy (CFL) criterion of limiting the time-step size during advection has a physical meaning, because it says that information from a given cell or mesh element must propagate only to its immediate neighbours.

**Reply:** Our time step controls outside the iterations (4.1.1 and 4.1.3) impact the physics like any other time stepping control, including the CFL criterion. The exception is the iteration-adaptive method (4.1.2) which is unstable. Likely because it interferes with the strain rate partitioning and stress state. We point this out in line 222 of the original manuscript.

* Adaptive rescaling (Section 4.2). The proposed scaling of variables seems ad-hoc - based on trial and error (L257). By non-dimensionalising the system of equations (rather than individual variables), one can derive non-dimensional numbers that control the

physics of the problem - i.e. elastic loading, creep and thermal run-away. Have the authors tried to take this approach instead to inform on the time step size? Alternatively, what are appropriate values of characteristic scales to capture correctly these processes? (L242) What is the impact of non-dimensionalisation? (L251).

**Reply:** There might be a misunderstanding about what we are describing in this section based around the terms nondimensionalization and characteristic scales. We will try to clarify this.

1) We are not looking to non-dimensionalize the equations to identify nondimensional numbers that describe the process of thermal runaway. This has been done in previous studies (Ogawa 1987, Braeck et al, 2009, Spang et al., 2024, all cited in the manuscript).

2) The sole purpose of internal scaling is to prevent round-off errors by centering quantities around 1 by dividing them by values which are typical for these quantities. This is stated in the first line of the section. It also has the side effect of making the quantities dimensionless.

3) This has no effect on the physical time step or any other physical quantity.

4) The adaptive rescaling is not an ad-hoc solution based on trial and error. In line 248 (now 283), we point out that the idea behind it is to prevent round-off errors due to numerical precision. Line 257 simply points out where this method starts to become effective. We went with $10^{-9}$ based on the numerical precision of $10^{-15}$, but the analysis shown in Figure 4 reveals that $10^{-12}$ is actually sufficient in our case.

To avoid the mix up with nondimensionalization of equations, we removed the term "characteristic" from the paragraph and renamed t_c to t_sc for scaling (same for other "characteristic scales"). We kept the subscript ND as the scaled quantities in the code are nondimensional on top of being scaled.

\* Melting during thermal runaway is not even discussed or considered as a potential way to limit runaway. Figures 5 and 6 show that Tmax > 5e3 deg C, in some cases reaching 1e5 deg C (L316). This is unrealistic, yet it is not even mentioned. The authors should at least discuss what would happen with the shear zone when temperatures go above the solidus and you have melting. It's possible that thermal diffusion (L37) and melting may limit the feedback loop.

**Reply:** Please see our reply on page 1 regarding the temperature and melting. Thermal diffusion is already in the model and cannot limit the runaway. The central concept of thermal runaway is that heat production by viscous dissipation is far greater than heat loss by thermal diffusion.

* Finally, how much of these problems are impacted by numerical errors due to the APT method (tolerance, iterations)? How about solving the system of equations with other methods such as Newton and Picard?

**Reply:** The APT method has numerical errors like every other method. Duretz et al., 2019 compare the APT method with a Newton solver and find no differences for the same problem of ductile strain localization due to shear heating. This paper is already cited in our section on the APT method. Testing different solver methods goes far beyond the scope of this study.

2. The writing requires revision and some re-structuring, mostly to avoid vague sentences and provide clarity.

* The manuscript contains challenge/challenging 18 times, which they seem to mean a number of things but never clear enough. Please reduce the usage and be more specific. Some examples (more in minor points),

**Reply:** Considering the title and topic of the manuscript, we don't think 18 is excessive. We also disagree that they mean a number of things. We always use this term to describe a situation that does not have an easy or obvious solution. You could replace every instance of "challenge" with "problem" and "challenging" with "difficult" and not change the meaning of the text. Except for line 332, we think all uses of these terms are appropriate. In most cases, these terms are used in the first sentence of a paragraph, and the following sentences give clarifications.

Section 4: "Challenges" could be renamed "Test cases"

**Reply:** We don't think that "Test cases" is a good description of the contents of Section 4. Section 4 shows the aspects of the model that are challenging (difficult, problematic) and how we solve them. We changed the title to "Numerical challenges and solution strategies".

L328-229: It would be helpful to tell the readers why it is challenging.

**Reply:** We specified that solution time for diffusive processes scales quadratically with the number of cells.

L332: what challenges?

**Reply:** We replaced "challenges" by "numerical errors". This is an introductory sentence; the problems are explained in the following paragraphs.

\* Structure and the logic of arguments in the introduction could be revised. First, strain localisation should be defined, either in the abstract (L1) or introduction (L15). Also clarify and explain the different types of localisations, brittle and ductile - and why is it difficult to solve them numerically. A parallel between brittle and ductile localisation would be helpful.

**Reply:** We agree with the reviewer that the introduction was missing a definition of strain localization. We have added a sentence to the beginning of the introduction to do that. We have also added another sentence to name a few geological processes which are governed by strain localization. The sentence in line 25 of the original manuscript has been extended by a short explanation of what brittle failure is. The start of the next paragraph (line 31 in the original manuscript) has been extended by an explanation of how ductile localization occurs and how it differs from brittle localization. We think lines 17-22 in the original manuscript already explain why localization is difficult to model.

\* The discussion could take the reader back to the introduction/motivation. Potentially propose other ways to regularise the model, in relation to the scientific problem (i.e., melting, grain size evolution or phase transitions for deep earthquakes; Billen et al. 2020). Also, what is the optimal recipe of the proposed methods in the end?

**Reply:** We added new subsections 6.2 and 6.3 to discuss grain size evolution and phase transformation. Melting is now part of the tested approaches. Regarding the recipe, in 1D, all methods (one out of viscosity and gradient regularization) are necessary. 2D is not as demanding due to the coarser resolution, this is now discussed in section 5.1 (Lines 380-393 in the original manuscript).

\* Methods section should have distinct parts for theory and numerical implementation. At the moment, they are mixed and it is confusing.
1. Theory: governing equations, constitutive equations such as Rheology (3.2) Density (3.3) - should be together.
2. Numerical implementation: APT method - including formulation for governing and constitutive equations, model setup (1D and 2D), boundary conditions, spatial discretisation.

**Reply:** Following the advice of reviewer 1, we restructured sections 2 and 3 in the following way: 2.1 Governing equations (including density now), 2.2 Rheology, 2.3 Model setup, 2.4 The 1D case. 3.1 Spatial discretization, 3.2 APT method, 3.3 Viscosity update, 3.4 Regularization.

L159: revise phrasing. For both the 1D and 2D models, we employ a variable grid, with the smallest cell size in the center of the model.

**Reply:** We rephrased the sentence as suggested.

* Section 4.3.3: Figures 5 and 6 should be combined (2 columns for each method, and 4 rows). Also, both figures should be discrete marker plots, not line plots - because experiments have not been done for a continuum of eta_reg and lambda_reg. The comparison between results should be clear this way.

**Reply:** We combined Figures 5 and 6 into one. We had to shrink the size of the individual panels to not push the caption off the page. We added dots to the lines to clarify which values we actually tested. Without lines, the Figures become difficult to read due to the many overlaps.

Minor points

L4: comes at a large computational cost

**Reply:** Fixed.

L5: result in rapid strain localisation (without "ductile")

**Reply:** We prefer to keep "ductile" here to emphasize that not all strain localization is brittle. We instead removed the "ductile" in front of "processes" in the same line to avoid repeating the word inside that sentence.

L6: delete "further"

**Reply:** Done.

L8: state which regularisation method are used

**Reply:** Done.

L13: rephrase "may differently impact..." being more precise.

**Reply:** We added another sentence after this one, explicitly stating which physical quantities are most affected by the two regularization approaches.

L15: rephrase: of solid deformation, common to solid deformable materials. Next sentence: how does strain localisation occur in solid Earth processes?

**Reply:** We included "solid" in the first (now second) sentence of the manuscript. We also added a follow-up sentence to mention where strain localization governs deformation.

L19: lacks a characteristic scale in time and space.

**Reply:** We would like to keep our phrasing.

L22: define plate-scale and grain-scale model.

**Reply:** We added the order of magnitude of the sizes of such models to the text.

L25: delete "However"

**Reply:** Done.

L28-30: disagree with this statement or at least how it is phrased: "strain localisation can occur at depths where ductile deformation would be expected".

**Reply:** We feel that this paragraph provides enough reasons for this statement as it is.

L34: Reference to Fig 2 - since both panels are referenced, remove "a,b".

**Reply:** Fixed.

L40: Summarise briefly the model in Spang et al (2024), i.e., using a 1D viscoelastic model ...

**Reply:** We are unsure what exactly the reviewer is asking here as the paragraph in question is already a brief summary of the model, its purpose, and behavior. We now added the word visco-elastic to the first sentence.

L47: the challenges to brittle failure have not been introduced before to say "similarly to brittle failure". Spiegelman et al (2016) is a good reference for brittle localisation.

**Reply:** We shifted this statement to be after the list of challenges and added the citation along with another appropriate one.

L50: what do the authors refer to as unstable solutions? non-reproducible?

**Reply:** We were referring to cases where the solver does not converge at all or might even diverge. As this is can also be summarized under poor solver convergence, we removed this point.

L53: reference for APT method?

**Reply:** Fixed.

L54-55: enumeration error: employing an adaptive time-stepping and monitoring viscosity convergence are not on the same level of action/strategy - one is active implementation, one is passive diagnostic.

**Reply:** As viscosity convergence is not only a monitored property but defines (among others) whether convergence is reached or not, we think that both are on a similar level. We adjusted the wording to "enforcing viscosity convergence".

L60, 64: the coefficients flow factors have not been introduced. This section should come after the theory.

**Reply:** This section was moved after the rheology section.

No need for section titles 2.2.1, 2.2.2. The reader will know from the text that you refer to 2D and a simplified 1D case.

**Reply:** Following the suggestions of Reviewer 1, these two sections were split apart and are now 2.1 and 2.4 respectively.

L69-70: this sentence is redundant.

**Reply:** We removed it.

L78: inertial and body forces from Eq. 1, thermal expansion and radiogenic heating from Eq. 3

**Reply:** Fixed.

Eq. 5: Are rotation and advection of stresses included in the elasticity term?

**Reply:** No, stresses are not advected or rotated. We added that we use the small strain approximation in the first sentence of section 3.

L86: Kronecker delta.

**Reply:** Fixed

L88: governing eps are only 1-3; Eq 1-6 are simplified such that…

**Reply:** Fixed.

L90-100: can be written in a compact form to follow Eqs. 1-6. At the moment, Eq 2, 4, 6, 1.

**Reply:** We agree, but we prefer the slightly longer version as we feel that it is easier to follow why all these simplifications can be made.

L136: it would be helpful to explain where each mechanism is expected to be dominant.

**Reply:** We added a sentence for that at the end of the Rheology section.

L146: cannot

**Reply:** Fixed.

L146: weird logic. The opposites are 1) analytical vs numerical; 2) direct vs iterative approach. Rephrase: cannot be solved analytically, such that it requires a numerical approach.

**Reply:** Fixed.

L146-153: should be in the APT section.

**Reply:** This part is now section 3.3 (Viscosity update) and follows the APT section.

Eq 27: why is density not a function of temperature as well? This could also help constrain the runaway. The choice of this constitutive equation should be better justified.

**Reply:** We tested thermal expansion in a previous 2D study (Spang et. al, 2025, cited in the manuscript) and found it to not play a significant role. We introduced a new section (6.1) to discuss the reasoning and/or effects of our simplifications. This section is also referenced after the governing equations.

L165: What is the minimum grid size for 2D?

**Reply:** 26 meters. We added the number of cells per direction and their respective size.

L166: use a staggered grid approach, where ....

**Reply:** Fixed.

L172: alternative regularisations for brittle failure

**Reply:** Fixed.

L191: Rephrase without "for conciseness": We use primarily 1D models to illustrate the challenges ... Similar problems arise in 2D models, which are discussed in Sect 5.

**Reply:** We now use the phrasing suggested by the reviewer.

L195: Evolution of model or description of problem is missing in Sec 4.1

**Reply:** The evolution of the model is described in the introduction and Figure 1b. We added a sentence to remind the reader of this. The problem is that we want large time steps for the loading and LTP stage and tiny time steps during runaway. This is what the first two sentences in the paragraph state.

L197: What is the duration of the runaway stage? How many time-steps are required?

**Reply:** As runaway is an exponential process, we find it difficult to define exactly where it starts and ends. If we define the start as the moment stress drops below 1800 MPa and the end as the moment where stress is 0, then the duration is about 250 years. If we define 1500 MPa as the start where the stress evolution appears to be vertical in Figure 3a, it is 5 years. If we define the runaway stage as the period with a stress change of more than 1 MPa/s (10% of the maximum stress gradient, covers the range between 1300 and 125 MPa), then it takes about 300 seconds. In any case, we are talking about 1500 to 1700 time steps for the example in Figure 3.

L201: vague statement. There are various of numerical methods proposed in the literature...

**Reply:** We are not sure if this comment is a suggestion for rephrasing or a rough quotation of the vague statement. The sentence in question acknowledges that our study is not the first to identify time stepping as an important problem and propose a solution. An exhaustive summary of available methods is beyond the scope of this section. We changed the sentence to "... and a number of studies propose different

methods" and added more references.

L205: what does it mean physically to exceed these values?

**Reply:** Exceeding these values means that the state of the system changes significantly within a single time step. This can have two consequences. 1) The solver does not converge. 2) Nonlinearities are underestimated and short-term processes are missed. Figure 3 shows exactly this. If the time step is too large, the feedback loop between shear heating and deformation is underestimated, and we see an incomplete runaway with much lower velocity and temperature.

Eq 31: what values are allowed for the min() term, when $dT_{n-1} < dT_{max}$? Should it be min(min(),1)? i.e., keep dt constant otherwise.

**Reply:** We allow an increase of up to 25%. So it would be min(..., 1.25). We pointed this out in line 235. But we agree that it is better to already state it here. We updated the equation.

L219: are model results for this case plotted in Fig. 3?

**Reply:** The method is unstable, so we did not plot it. We added a sentence to clarify this.

L223: plot solver behaviour? If solutions are unreliable, this is not a good method.

**Reply:** It is indeed not a good method. We added a sentence to clarify that. But we still felt that it was worth mentioning that we tested it in case a reader might consider using it.

L223: define PT iterations, or use APT

**Reply:** PT is defined in line 149 of the original manuscript.

L245, Eq (32): this means a non-dimensional time step (remove "would internally be") and the equation doesn't need to be separate line.

**Reply:** Fixed.

L254: two orders magnitude

**Reply:** Fixed.

L261: quantify dramatically. i.e. up to 10 orders magnitude.

**Reply:** We added that it decreases more than 10 orders of magnitude. How much exactly, depends on the regularization.

L262: generally challenging...? What does it mean?

**Reply:** Challenging means difficult. We removed "generally".

L264: instead of the physics of the problem.

**Reply:** We adopted this phrasing.

L267: 60 1-D simulations? More explicit about what numerical resolutions and values of eta_reg are used.

**Reply:** We added the values for all three parameters.

L275: statement not clear: As eta_reg is further reduced, this divergence propagates to higher-resolution models, following the same pattern.

**Reply:** We rephrased this sentence for clarification.

L288-289: repeated sentence as in previous section. It's better to state at the beginning: the following diagnostics are used...

**Reply:** We moved the list of diagnostic parameters to the top paragraph of the regularization section to avoid repeating it. The sentence structure is still repeated in sections 4.3.1 and 4.3.2 but we do not think that this hurts the flow of the text. Instead, it illustrates that both methods have the same first-order effect.

L297-298: What was the stopping criterion for these simulations? Why sims with high lambda_reg didn't finish?

**Reply:** The stopping criterion was the allocated run time of one day. We continued these simulations from a restart database, but most of them did not finish after a second day either, due to extremely small timesteps. We considered continuing them a waste of computational resources as in our opinion the trend can be observed in Figure 6d as it is.

The reason for the slow convergence is that the gradient regularization diffuses heat (or shear heating to be more precise) and the iteration count for diffusive processes scales quadratically with resolution. At the same time, increasing the diffusion coefficient

(lambda_reg) reduces the physical time step, thus more time steps are required for the same time period. This is pointed out in section 4.3.3 (now 4.3.4).

L308: known correspondence between eta_reg and lambda_reg.

**Reply:** Fixed.

L340: This statement on LTP should come earlier where rheology is introduced.

**Reply:** We copied this statement to the rheology section and shortened it here.

L395: reproducibility is also a problem.

**Reply:** Yes, reproducibility is a problem. Regularization is a powerful tool to make numerical results more reproducible. We now mention reproducibility in the introduction and throughout the manuscript.

Figures and Tables

Figure 1: explain briefly why LTP is dominant in the stage prior to runaway but dislocation creep is dominant in the stage post runaway.

**Reply:** We instead added an explanation to the text section that references Figure 1b. (Line 57 in the new manuscript).

Figure 2: What are the height and length of models?

**Reply:** We added the extents of the models to the caption. They are also given in the model setup section.

Table 1: There should be another column explaining the physical meaning of each parameter.

**Reply:** We added such a column.

Figure 3: Revise caption: Model results with fixed and adaptive time-stepping. Panel b (or new panel): Plot T increase and viscosity decrease. Panel a: there is a large stress difference post runaway between methods. What causes that?

**Reply:** We adapted the Figure title and added panels for the evolution of temperature, viscosity, and slip velocity. The evolution of timestep was turned into an inset. The larger

time steps result in less violent runaway (i.e., less localization, less temperature increase). Therefore, the viscosity remains larger and stress is not fully released. As those cases don't get as hot, the shear zone can cool down enough to rebuild stress.

Figure 4: Caption: Model results using adaptive rescaling. Sum of iterations or number of iterations? Comma, not full-stop before eta_reg.

**Reply:** We think that "Effect of adaptive rescaling" is a better description of what the Figure shows. The Figure shows the sum of iterations. We kept the full stop in front of eta_reg as it is not a model result like the others listed.

Figure 5: a-b) Legend should be the same. The corresponding link between number of cells and min grid size should be explained in the main text (e.g., L267). These plots should not have continuous lines, they are discrete data points for various eta_reg (same for figure 6).

**Reply:** We think the two legends are very helpful. We now mention the correspondence in the caption. It is already mentioned in the text in line 273 of the original manuscript. We added dots to the lines to clarify which values we tested. Without lines, the Figures become difficult to read due to the many overlaps. As requested above, we combined Figures 5 and 6 into one.

Figure 7: The tolerance was not defined or introduced, i.e. solver tolerance? Panel b is confusing, and not clear what this figure tries to show.

**Reply:** We fixed this oversight. We now introduce the viscosity tolerance in the text before the Figure. Panel b shows how the viscosity $eta^{it}$ evolves during the PT iterations towards a target viscosity $eta^t$ according to the viscosity relaxation method (Eq. 27, 26 in the original manuscript). It illustrates that this takes hundreds or thousands of PT iterations, depending on the relaxation parameter eta_rel. This should help explain why the stress overshoots in (a) can occur when the convergence between $eta^{it}$ and $eta^t$ is not part of the convergence criterion. We made some adjustments in the text and replaced "start" and "target" by $eta^{it}$ and $eta^t$ which should clarify this.

Figure 8: caption should have the reference parameters stated and the regularisation methods employed.

**Reply:** We added this information to the caption.

Figure 9: reference parameters need to be stated for both 1d and 2D runs.

**Reply:** We added this information to the caption.

Code - Readme.md

1. Provide installation instructions and dependencies. Example errors encountered:

ERROR: LoadError: ArgumentError: Package ParallelStencil not found in current path.

ERROR: LoadError: ArgumentError: Package Plots not found in current path.

**Reply:** We now explain two different ways to install the required packages.

2. Explain function inputs: ElaDisDifLTP_1D(5e-13, 80.0, 1.8, 10.0, 3.0, 600.0, 2.0, 0.02, 1e15, 127, "Ref")

**Reply:** Done.